# Inverse Problem Sampling in Latent Space Using Sequential Monte Carlo

**Idan Achituve** [1]   **Hai Victor Habi** [1]   **Amir Rosenfeld** [1]   **Arnon Netzer** [1]   **Idit Diamant** [* 1]   **Ethan Fetaya** [* 2]

## Abstract

In image processing, solving inverse problems is the task of finding plausible reconstructions of an image that was corrupted by some (usually known) degradation operator. Commonly, this process is done using a generative image model that can guide the reconstruction towards solutions that appear natural. The success of diffusion models over the last few years has made them a leading candidate for this task. However, the sequential nature of diffusion models makes this conditional sampling process challenging. Furthermore, since diffusion models are often defined in the latent space of an autoencoder, the encoder-decoder transformations introduce additional difficulties. To address these challenges, we suggest a novel sampling method based on sequential Monte Carlo (SMC) in the latent space of diffusion models. We name our method LD-SMC. We define a generative model for the data using additional auxiliary observations and perform posterior inference with SMC sampling based on a reverse diffusion process. Empirical evaluations on ImageNet and FFHQ show the benefits of LD-SMC over competing methods in various inverse problem tasks and especially in challenging inpainting tasks.

## 1. Introduction

Many important signal processing tasks can be viewed as inverse problems (Song et al., 2021c; Moliner et al., 2023; Daras et al., 2024; Chung et al., 2023b; Cardoso et al., 2023). In inverse problems, the objective is to obtain a clean signal $\mathbf{x} \in \mathbb{R}^n$ from a degraded observation $\mathbf{y} = \mathcal{A}(\mathbf{x}) + \psi$, where $\mathcal{A}$ is usually a known irreversible mapping and $\psi$ is a Gaussian noise vector. Common applications that fit this framework include image deblurring, super-resolution, inpainting,

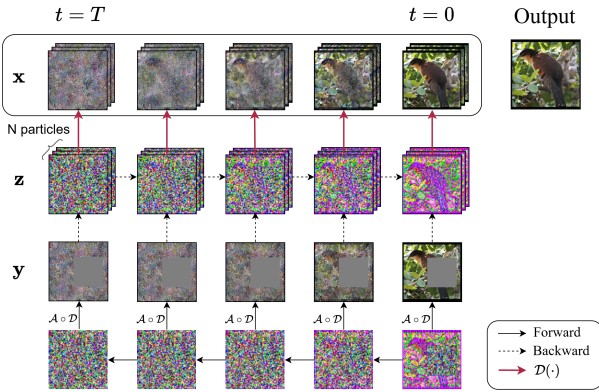

Figure 1: LD-SMC solves inverse problem tasks in the latent space of autoencoders by utilizing auxiliary observations $\mathbf{y}_{1:T}$ initialized using the DDIM forward process. Then, sampling $\mathbf{z}_0$ from the posterior distribution is done based on the reverse diffusion process using sequential Monte Carlo. In the figure, $\mathcal{D}$ and $\mathcal{A}$ denote the decoder and the corruption operator respectively.

and Gaussian denoising. The broad applicability of inverse problems makes them highly significant, as they encompass numerous real-world challenges, such as those found in digital image processing (Blackledge, 2005), wireless communication (Chen et al., 2021), seismology (Virieux & Operto, 2009), medical imaging (Song et al., 2021c; Chung et al., 2023c), and astronomy (Craig & Brown, 1986).

A major challenge in solving inverse problems is the existence of multiple plausible solutions. For example, in image inpainting, the likelihood $p(\mathbf{y}|\mathbf{x})$ remains constant regardless of how the absent pixels are filled. However, the desired solution is one that not only fits the observation, but also appears natural, which corresponds to having a high probability under a natural image prior $p(\mathbf{x})$. This insight naturally leads to the approach of sampling from the posterior distribution $p(\mathbf{x}|\mathbf{y}) \propto p(\mathbf{y}|\mathbf{x})p(\mathbf{x})$, combining the data likelihood and the prior to achieve realistic and data-consistent solutions.

With the impressive recent advances in diffusion models (Sohl-Dickstein et al., 2015; Ho et al., 2020; Song et al., 2021a), there has been a significant interest in leveraging

---

*Equal contribution  [1] Sony Semiconductor Israel (SSI), Israel [2]Faculty of Engineering, Bar-Ilan University, Israel. Correspondence to: Idan Achituve <Idanachi@gmail.com>.

*Proceedings of the 42nd International Conference on Machine Learning*, Vancouver, Canada. PMLR 267, 2025. Copyright 2025 by the author(s).

them as prior image models to solve inverse problems. However, integrating diffusion models into this context is not straightforward because of their sequential sampling process. Specifically, diffusion sampling involves iterative drawing from $p(\mathbf{x}_{t-1}|\mathbf{x}_t)$, while the conditioning on the corrupted image $\mathbf{y}$ is defined only in the final step, namely, through $p(\mathbf{y}|\mathbf{x}_0)$. This mismatch makes direct sampling from the joint posterior $p(\mathbf{x}_0, ..., \mathbf{x}_T|\mathbf{y})$ particularly challenging.

Recently, several studies proposed sequential Monte Carlo (SMC) (Doucet et al., 2001b; Del Moral et al., 2012) as an effective solution for this task based on pixel-space diffusion models (Cardoso et al., 2023; Trippe et al., 2023; Wu et al., 2023; Dou & Song, 2024). Specifically, Wu et al. (2023) applies the DPS approximation (Chung et al., 2023b) for $p(\mathbf{y}|\mathbf{x}_t)$ with $p(\mathbf{y}|\mathbb{E}[\mathbf{x}_0|\mathbf{x}_t])$ and uses SMC sampling to correct for it. Dou & Song (2024), on the other hand, connected $\mathbf{x}_t$ to $\mathbf{y}$ by introducing a sequence of latent variables $\mathbf{y}_{1:T}$ through a duplex forward diffusion process and sampling sequentially from $p(\mathbf{x}_{t:T}|\mathbf{y}_{t:T})$. While this approach has shown great potential, it has two main limitations. First, it does not take into account future observations $\mathbf{y}_{0:t-1}$ in the sampling process. Second, and more crucially, it is limited to linear corruption models only. As such, it cannot be applied with nonlinear mappings $\mathcal{A}$, let alone common Latent Diffusion Models (LDMs) (Rombach et al., 2022) due to the nonlinearity of the decoder. This is a harsh restriction as many of the recent powerful and efficient models are LDMs (Esser et al., 2024).

Both existing approaches have pros and cons. Using the $p(\mathbf{y}|\mathbb{E}[\mathbf{x}_0|\mathbf{x}_t])$ approximation can be helpful in capturing the large-scale semantics of the image, but it often lacks in capturing the small details. On the other hand, using auxiliary observations $\mathbf{y}_{1:T}$ can help capture finer details, but the duplex forward diffusion process is not amenable to LDMs. Here, we propose a method that combines these two approaches and strives to achieve the best of both worlds. We define a generative model for the data based on a reverse diffusion process, according to which an auxiliary observation $\mathbf{y}_t$ is generated directly from $\mathbf{z}_t$, the latent diffusion variable. Then, to apply posterior inference over the variables of all timesteps, $\mathbf{z}_{0:T}$, we use SMC. To obtain a tractable sampling procedure, we derive an approximate target distributions and define a novel proposal distributions for the SMC sampling process tailored for diffusion models. Hence, we name our method Latent Diffusion Sequential Monte Carlo, or more concisely LD-SMC. An illustration of our approach is shown in Figure 1. We empirically validated LD-SMC on the ImageNet (Russakovsky et al., 2015) and FFHQ (Karras et al., 2019) datasets. We found that LD-SMC outperforms or is comparable to baseline methods on image deblurring and super-resolution tasks, and can significantly improve over baseline methods on inapainting tasks, especially on the more diverse ImageNet dataset.

In this study, we make the following contributions: (1) we propose a novel method for combining auxiliary observations with latent space diffusion models; (2) we derive approximate target distributions for the SMC procedure and novel proposal distributions to perform approximate posterior sampling specifically tailored for diffusion models; (3) We theoretically show that LD-SMC is asymptotically accurate, namely that it can sample from $p_\theta(\mathbf{z}_0|\mathbf{y}_0)$; (4) LD-SMC achieves significant improvements in perceptual quality in inpainting tasks, one of the most challenging inverse problem tasks.

## 2. Background

**Inverse Problems.** In inverse problems, one would like to recover a sample $\mathbf{x} \in \mathbb{R}^n$ from a corrupted version of it $\mathbf{y} \in \mathbb{R}^m$. Usually, the corruption model that acted on $\mathbf{x}$ is known, but the operation is irreversible (Tarantola, 2005). For instance, restoring a high-quality image from a low-quality one. We denote the corruption operator by $\mathcal{A}(\cdot)$, and assume that $\mathbf{y} = \mathcal{A}(\mathbf{x}) + \boldsymbol{\psi}$, where $\boldsymbol{\psi} \sim \mathcal{N}(0, \tau^2\mathbf{I})$ has a known standard deviation $\tau$. In a more concise way, $p(\mathbf{y}|\mathbf{x}) = \mathcal{N}(\mathcal{A}(\mathbf{x}), \tau^2\mathbf{I})$. Common examples of inverse problems are inpainting, colorization, and deblurring. In general, solving inverse problem tasks is considered an ill-posed problem with many possible solutions $\mathbf{x}$ with equally high $p(\mathbf{y}|\mathbf{x})$ values. Given a prior distribution $p(\mathbf{x})$ over natural images, one standard approach to solving the inverse problem is to sample from the posterior distribution $p(\mathbf{x}|\mathbf{y}) \propto p(\mathbf{y}|\mathbf{x})p(\mathbf{x})$.

**Diffusion Models.** Owing to their high-quality generation capabilities, in recent years diffusion models (Sohl-Dickstein et al., 2015; Ho et al., 2020) have been leveraged as priors in inverse problems (Jalal et al., 2021; Song et al., 2021d). Here, we adopt the DDIM formulation (Song et al., 2021a) for the prior model, although our approach can work with other formulations of diffusion models as well. Furthermore, since it is costly to apply the diffusion process in the pixel space, a common approach is to apply the diffusion model in the latent space given by an auto-encoder (Rombach et al., 2022). Applying diffusion models in latent space allows us to sample high-quality images while reducing the computational resources needed by the model. Hence, designing models that effectively solve inverse problems using latent diffusion models is of great importance.

Denote by $\mathbf{z}_{0:T}$ the random variables in the latent space. Let $\alpha_{1:T}, \beta_{1:T}$ be the variance schedule of the diffusion process with $\beta_t := 1 - \alpha_t$. Also, denote by $\bar{\alpha}_t := \prod_{j=1}^{t} \alpha_j$. The DDIM sampling starts from a prior distribution at timestep $T$ set to $p(\mathbf{z}_T) = \mathcal{N}(0, \mathbf{I})$. Then, for $t > 1$ the sampling is done according to $p_\theta(\mathbf{z}_{t-1}|\mathbf{z}_t) = \mathcal{N}(\mathbf{z}_{t-1}|\boldsymbol{\mu}_\theta(\mathbf{z}_t, t), \boldsymbol{\Sigma}(t))$,

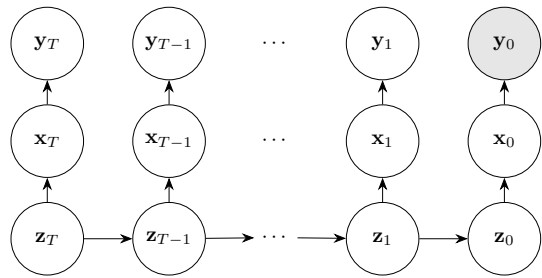

Figure 2: The graphical model of LD-SMC. In gray observed variables and in white are latent variables.

where $\theta$ are the parameters of the neural network and,

$$\boldsymbol{\Sigma}(t) = \sigma_t^2 \mathbf{I}$$

$$\boldsymbol{\mu}_\theta(\mathbf{z}_t, t) = \sqrt{\bar{\alpha}_{t-1}} \bar{\mathbf{z}}_0(\mathbf{z}_t) + \sqrt{1 - \bar{\alpha}_{t-1} - \sigma_t^2} \cdot \boldsymbol{\epsilon}_\theta(\mathbf{z}_t, t). \quad (1)$$

Here we denote the approximate posterior mean of $\mathbb{E}[\mathbf{z}_0|\mathbf{z}_t]$ by $\bar{\mathbf{z}}_0(\mathbf{z}_t) \coloneqq \frac{1}{\sqrt{\bar{\alpha}_t}}(\mathbf{z}_t - \sqrt{1 - \bar{\alpha}_t} \cdot \boldsymbol{\epsilon}_\theta(\mathbf{z}_t, t))$ (Robbins, 1956; Efron, 2011; Chung et al., 2023b). For timestep $t = 1$, the mean $\boldsymbol{\mu}_\theta(\mathbf{z}_1, 1)$ is taken to be $\bar{\mathbf{z}}_0(\mathbf{z}_1)$. As in (Dou & Song, 2024) we fix $\sigma_t = \eta \cdot \sqrt{\beta_t \cdot \frac{1 - \bar{\alpha}_{t-1}}{1 - \bar{\alpha}_t}}$ with $\eta$ being a hyperparameter.

**Sequential Monte Carlo (SMC).** SMC is an important technique for sampling in probabilistic graphical models in which exact posterior inference is intractable. The SMC breaks the sampling process down to intermediate steps, allowing efficient sampling through a recursive procedure (Doucet et al., 2001b; Del Moral et al., 2012; Naesseth et al., 2019; Chopin et al., 2020).

One family of probabilistic models for which SMC is especially known is state-space models (SSMs), also known as Hidden Markov Models (HMMs). In general, the following quantities must be defined in SSMs, (1) a prior distribution over the initial state $p(\mathbf{z}_T)$, (2) a transition distribution that defines the dynamics between states $p(\mathbf{z}_t|\mathbf{z}_{t+1}) \; \forall t < T$, and (3) a measurement model $p(\mathbf{y}_t|\mathbf{z}_t) \; \forall t < T$. The goal is to sample from the target distribution $p(\mathbf{z}_{t:T}|\mathbf{y}_{t:T-1})$. To do so, SMC starts by sampling $N$ particles $\{\mathbf{z}_T^{(i)}\}_{i=1}^N$ from the prior distribution. Then, at each step, given the previous particle set $\{\mathbf{z}_t^{(i)}\}_{i=1}^N$ new samples are taken from a proposal distribution $\mathbf{z}_{t-1}^{(i)} \sim \pi_{t-1}(\mathbf{z}_{t-1}^{(i)}|\mathbf{z}_t^{(i)}) \; \forall i \in \{1, ..., N\}$. The particles are then weighted (and possibly resampled) according to the new sequences $\{\mathbf{z}_{t-1:T}^{(i)}\}_{i=1}^N$. The proposal distribution serves as an approximation to the target distribution. Its support needs to contain the support of the target density. The weighting mechanism then corrects the approximation by assigning a weight to each particle to adjust its probability. The resampling step, if applied, aims to remove unlikely particles according to the model (Särkkä, 2013).

## 3. Related Work

Inverse problems have a long and evolving history, with methodologies that have undergone significant advances over the years (Daras et al., 2024). Recently, diffusion models (Sohl-Dickstein et al., 2015; Ho et al., 2020; Song et al., 2021d) have emerged as effective priors for solving inverse problems in image data (Choi et al., 2021; Kawar et al., 2022; Chung et al., 2023a;b; Rout et al., 2023; Song et al., 2023; Wang et al., 2023; Dou & Song, 2024; Garber & Tirer, 2024; Mardani et al., 2024; Sun et al., 2024).

In (Song et al., 2021d) it was shown that to sample from the posterior distribution, $p(\mathbf{x}_0|\mathbf{y})$, one can solve a stochastic differential equation based on the prior score, $\nabla_{\mathbf{x}_t} \log p_\theta(\mathbf{x}_t)$, and the conditional score, $\nabla_{\mathbf{x}_t} \log p_\theta(\mathbf{y}|\mathbf{x}_t)$. While the first term is easy to compute, the latter term requires integration over the full diffusion path from time $t$ to 0. A useful and easy-to-calculate approximation found in several studies is $p_\theta(\mathbf{y}|\mathbf{x}_t) \approx p_\theta(\mathbf{y}|\mathbb{E}[\mathbf{x}_0|\mathbf{x}_t]) \approx p_\theta(\mathbf{y}|\bar{\mathbf{x}}_0(\mathbf{x}_t))$, which is readily available at each step (Chung et al., 2023b; Song et al., 2023; Wu et al., 2023). Specifically, Diffusion Posterior Sampling (DPS) (Chung et al., 2023b) uses this approximation for linear and nonlinear inverse problems with Gaussian and Poisson likelihood models. Other methods also utilize the pseudoinverse of the corruption operator $\mathcal{A}$ (Tirer & Giryes, 2018). ΠGDM (Song et al., 2023) introduces a guidance scheme by matching the denoised output and the corrupted image $\mathbf{y}$, via a transformation of both. DDNM (Wang et al., 2023) suggested refining the contents of the null space of $\mathcal{A}$ during the reverse diffusion process. A different approach proposed in (Mardani et al., 2024) is to approximate the posterior using a variational approach based on the score matching objective presented in (Song et al., 2021b). An additional category of inverse problem approaches that use diffusion models is designed with the objective of asymptotic exactness (Cardoso et al., 2023; Trippe et al., 2023; Wu et al., 2023; Dou & Song, 2024). These methods utilize SMC techniques, which have also been applied for unconditional sampling (Chen et al., 2025), to target exact sampling from the posterior distribution $p(\mathbf{x}_0|\mathbf{y})$. SMC-Diff (Trippe et al., 2023) was designed mainly for motif-scaffolding. It uses the prior distribution as a proposal which is known to require a large number of particles for accurate estimation results (Särkkä, 2013), a severe limitation for latent space diffusion models due to expensive decoder evaluations. We experimented with this proposal in our initial attempts and witnessed that even 150 particles were not enough to sample plausible reconstructions. MCGDiff (Cardoso et al., 2023) was designed for linear inverse problems only, which makes it unsuitable for inverse problems with latent-space diffusion models due to encoder-decoder involvement. TDS (Wu et al., 2023), a recent SMC-based method, solves general inverse problem tasks using the twisting technique. This

**Algorithm 1** LD-SMC

---

1: Set $\mathbf{z}_0 = \mathcal{E}(\mathbf{x}_0)$, where $\mathbf{x}_0 = \arg\min_{\mathbf{x}} ||\mathbf{y}_0 - \mathcal{A}(\mathbf{x})||_2^2$
2: Sample $\mathbf{z}_{1:T} \sim p(\mathbf{z}_{1:T}|\mathbf{z}_0)$ according to the forward process of DDIM (Eq. 2)
3: **For** $k = 1, ..., K$:
4:     Sample $\mathbf{y}_{1:T} \sim \prod_{t=1}^{T} \mathcal{N}(\mathbf{y}_t|\mathcal{A}(\mathcal{D}(\mathbf{z}_t)), \tau^2\mathbf{I})$
5:     Sample $\mathbf{z}_{0:T} \sim p_\theta(\mathbf{z}_{0:T}|\mathbf{y}_{0:T})$ using **SISR**$(\mathbf{y}_{0:T})$

6: **SISR**$(\mathbf{y}_{0:T})$:
7:     **For** $i = 1, ..., N$:
8:       Sample $\mathbf{z}_T^{(i)} \sim p_\theta(\mathbf{z}_T)$ and initialize $\mathbf{z}^{(i)} = (\mathbf{z}_T^{(i)})$
9:       Set $w_T^{(i)}$ according to Eq. 6
10:    Normalize $\{w_T^{(i)}\}_{i=1}^N$ to sum to one
11:    **For** $t = T - 1, ..., 0$:
12:      $\{\mathbf{z}^{(i)}\}_{i=1}^N \sim \text{Multi}(\{\mathbf{z}^{(i)}\}_{i=1}^N, \{w_{t+1}^{(i)}\}_{i=1}^N)$, a resampling step
13:      **For** $i = 1, ..., N$:
14:        Sample $\mathbf{z}_t^{(i)} \sim \pi_t(\mathbf{z}_t|\mathbf{z}_{t+1}^{(i)})$ (Eq. 7) # proposal
15:        Set $w_t^{(i)}$ according to Eq. 6
16:        Update $\mathbf{z}^{(i)} = (\mathbf{z}_t^{(i)}, ..., \mathbf{z}_T^{(i)})$
17:      Normalize $\{w_t^{(i)}\}_{i=1}^N$ to sum to one
18:    Sample one chain $\mathbf{z} \sim \text{Multi}(\{\mathbf{z}^{(i)}\}_{i=1}^N, \{w_0^{(i)}\}_{i=1}^N)$
19:    **Return z**

---

method also uses the approximation of DPS, but by applying SMC sampling it can correct for it. FPS (Dou & Song, 2024), also a recent SMC-based method, uses auxiliary variables. FPS generates a sequence of observations $\mathbf{y}_{1:T}$ based on a duplex diffusion process, one process at the $\mathbf{x}$ space and the other process at the $\mathbf{y}$ space. Since this method is designed for linear inverse problems only, it permits tractable Bayesian inference. Our method combines the ideas of both TDS and FPS to obtain the best of both. Namely, we use the posterior mean approximation and $\mathbf{y}_{1:T}$ in our SMC sampling process. As we will show, this combination can be helpful in both understanding the general semantics of an image and capturing fine details. We note here that there are other methods that applied label augmentations similar to FPS, one example is (Abu-Hussein et al., 2022) which is applicable for linear inverse problems only. Hence, this method as well cannot be trivially combined with latent space diffusion models.

Several inverse sampling methods were specifically tailored for latent diffusion models. PSLD (Rout et al., 2023) extend DPS (Chung et al., 2023b) by incorporating an additional gradient update step to guide the diffusion process to sample latent representations that maintain the integrity of the decoding-encoding transformation, ensuring it remains non-lossy. STSL (Rout et al., 2024) uses second-order information for reconstruction based on efficient approximations. Comparative analysis with STSL was not feasible due to

the absence of publicly available code, making replication challenging. ReSample (Song et al., 2024), a contemporary method alongside PSLD, acts in two stages; it first applies hard data consistency to obtain latent variables that are consistent with the observed measurements and then employs a resampling scheme to map the sample back onto the data manifold. Concurrent to this study Nazemi et al. (2024) proposed a particle filtering approach. Their method builds on PSLD and DPS update in the proposal distribution. Similarly to TDS (Wu et al., 2023) the connection to the labels is only through $\mathbf{z}_0$ using the approximate mean estimator. Also, unlike LD-SMC it does not enjoy asymptotic guarantees. Recently, both DAPS (Zhang et al., 2024) and MGPS (Moufad et al., 2024) have been proposed to address inverse problems both in pixel and latent space. DAPS decouples latents in the sampling trajectory by first sampling from $\mathbf{z}_0|\mathbf{z}_{t+\Delta t}, \mathbf{y}$ using MCMC techniques, for some $\Delta t > 0$, and then sampling $\mathbf{z}_t$. MGPS also decomposes the sampling, but to an intermediate midpoint and uses variational inference for posterior sampling. Given the similarities of both methods, in our experiments we compare only to the former. SILO (Raphaeli et al., 2025), an additional recent study, first learns a NN that mimics an encoder operation on labels $\mathbf{y}$ which is then used to guide the sampling process. In our empirical evaluation, we compare to methods that apply sampling only.

## 4. Method

Given a corrupted image $\mathbf{y}_0$, the goal is to sample $\mathbf{z}_0 \sim p_\theta(\mathbf{z}_0|\mathbf{y}_0)$ using a pre-trained latent diffusion model as prior. Then, we can transform this sample into an image by applying a pre-trained decoder $\mathcal{D}$, i.e. $\mathbf{x}_0 := \mathcal{D}(\mathbf{z}_0)$. Here, we stack all the parameters of the networks (diffusion and encoder-decoder) under $\theta$. In what follows, we first define a generative model for the data and then describe how to perform Bayesian inference on all latent variables using blocked Gibbs sampling. Specifically, we use the diffusion process and the corruption operator to augment the model with additional auxiliary observations. Then, inference is applied over the diffusion variables using SMC. The corresponding graphical model can be seen in Figure 2.

### 4.1. The Generative Model

We now explicitly define the data generation model,

    1. $\mathbf{z}_T \sim \mathcal{N}(0, \mathbf{I})$,
    2. $\mathbf{z}_{t-1}|\mathbf{z}_t \sim p(\mathbf{z}_{t-1}|\mathbf{z}_t) \quad \forall t \in \{1, ..., T\}$,
    3. $\mathbf{y}_t|\mathbf{z}_t \sim \mathcal{N}(\mathcal{A}(\underbrace{\mathcal{D}(\mathbf{z}_t)}_{\mathbf{x}_t}), \tau^2\mathbf{I}) \quad \forall t \in \{0, ..., T\}$.

Where, $p_\theta(\mathbf{y}_t|\mathbf{z}_t)$ is defined by the corruption model, $p(\mathbf{z}_{t-1}|\mathbf{z}_t) = \mathbb{E}_{p(\mathbf{z}_0|\mathbf{z}_t)}[p(\mathbf{z}_{t-1}|\mathbf{z}_t, \mathbf{z}_0)]$ is a backwards generative process that corresponds to a non-Markovian forward

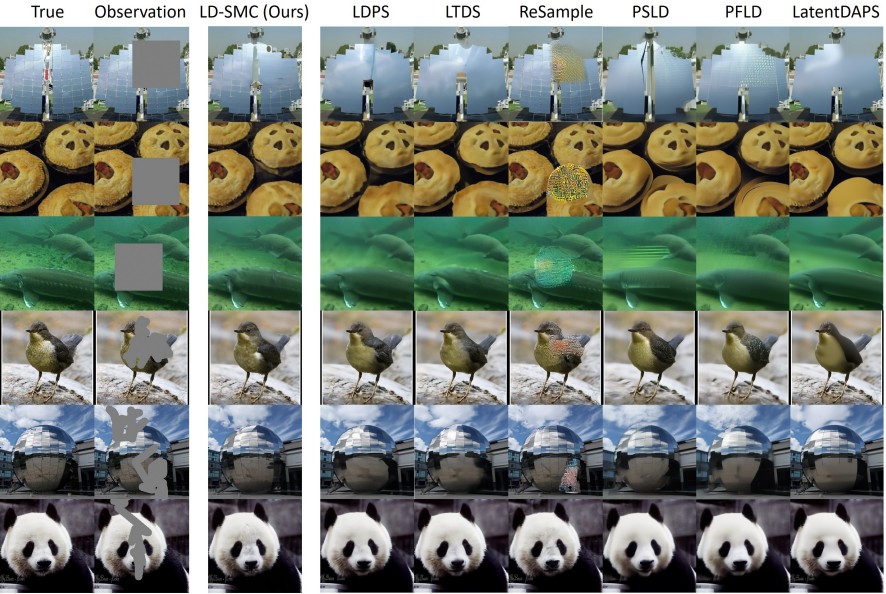

Figure 3: Comparison between LD-SMC and baseline methods on inpainting of ImageNet images.

process with

$$p(\mathbf{z}_{t-1}|\mathbf{z}_t, \mathbf{z}_0) =$$

$$\mathcal{N}\left(\mathbf{z}_{t-1}|\sqrt{\bar{\alpha}_{t-1}}\mathbf{z}_0 + \sqrt{1 - \bar{\alpha}_{t-1} - \sigma_t^2} \cdot \frac{\mathbf{z}_t - \sqrt{\bar{\alpha}_t}\mathbf{z}_0}{\sqrt{1 - \bar{\alpha}_t}}, \sigma_t^2\mathbf{I}\right),$$

(2)

set according DDIM (Song et al., 2021a). Introducing unobserved data is a known technique in statistics for conducting effective Markov chain Monte Carlo (MCMC) sampling (Van Dyk & Meng, 2001; Dou & Song, 2024). In our case, we can use it while leveraging the dependencies between the variables in order to build an efficient SMC sampling procedure, as described in the next section.

### 4.2. Sampling Procedure

Given the generative model defined in Section 4.1, our aim is to apply Bayesian inference over the latent variables. In broad strokes, to obtain a sample $\mathbf{z}_0 \sim p_\theta(\mathbf{z}_0|\mathbf{y}_0)$ we use blocked Gibbs sampling to sample in turns from $p_\theta(\mathbf{y}_{1:T}|\mathbf{z}_{0:T}, \mathbf{y}_0)$ using knowledge on the corruption model and $p_\theta(\mathbf{z}_{0:T}|\mathbf{y}_{0:T})$ using SMC. Specifically, we propose the following procedure:

1. Obtain an initial guess for $\mathbf{z}_0$ (Sec. 4.2.1).

2. Sample, $\mathbf{z}_{1:T} \sim p(\mathbf{z}_{1:T}|\mathbf{z}_0, \mathbf{y}_0) = p(\mathbf{z}_{1:T}|\mathbf{z}_0)$ according to the forward process of DDIM (Eq. 2).

3. Repeat for some fixed number of steps:

   (a) Sample, $\mathbf{y}_{1:T} \sim p_\theta(\mathbf{y}_{1:T}|\mathbf{z}_{0:T}, \mathbf{y}_0) = p_\theta(\mathbf{y}_{1:T}|\mathbf{z}_{1:T}) = \prod_{t=1}^{T} \mathcal{N}(\mathbf{y}_t|\mathcal{A}(\mathcal{D}(\mathbf{z}_t)), \tau^2\mathbf{I})$.

   (b) Sample, $\mathbf{z}_{0:T} \sim p_\theta(\mathbf{z}_{0:T}|\mathbf{y}_{0:T})$ using SMC based on a pre-trained diffusion model (Sec. 4.2.2).

Here, we use the dynamics of the forward process and the graphical model dependencies in steps 2 and 3(a). The two steps that are not straightforward are obtaining an initial guess for $\mathbf{z}_0$ (step 1) and performing the sampling process in step 3(b). We discuss both in the following sections, but before that we make a few comments. First, clearly the run-time of the algorithm depends linearly on the number of Gibbs iterations. As a result, the sampling time can be slow. However, with proper initialization (as discussed in Section 4.2.1), we empirically found that even one iteration of Step 3 suffices to achieve good results. Hence, unless otherwise stated, in our empirical evaluations, we applied this step only once. Second, one may be concerned that evaluating the decoder on noisy latent variables can generate non-natural images. While this may be true, empirically we observed a graceful degradation in image quality with time, as seen in Figure 7. Furthermore, the latent variables, even if noisy, carry information about $\mathbf{y}_0$. This information is then transferred to the auxiliary labels which help guide the sampling process.

#### 4.2.1. INITIAL GUESS FOR $\mathbf{z}_0$

The first challenge is to obtain an initial $\mathbf{z}_0$ (step 1. in the sampling procedure). There are multiple sensible ways to perform this initialization. Importantly, $\mathbf{z}_0$ should carry information about the measurement $\mathbf{y}_0$, which can then be used to guide the sampling process using the auxiliary la-

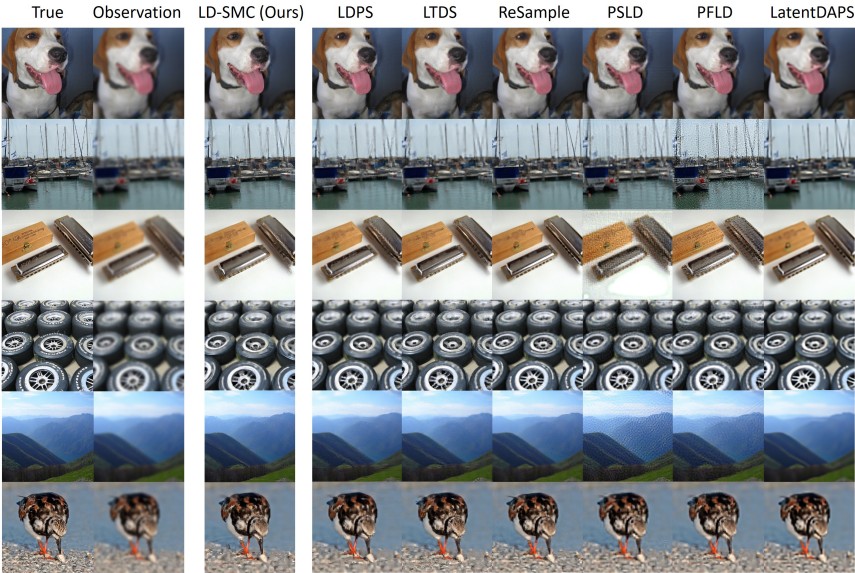

True Observation LD-SMC (Ours) LDPS LTDS ReSample PSLD PFLD LatentDAPS

Figure 4: Comparison between LD-SMC and baseline methods on Gaussian deblurring of ImageNet images.

bels. As a result, the variance in the process will be reduced and the convergence will be accelerated. For example, one option is to leverage the pseudoinverse of $\mathcal{A}$ for linear operators similar to (Wang et al., 2023). But, that would not work for nonlinear corruption operators. Although there are mitigations for this problem (Song et al., 2023), to maintain generality, we applied the following optimization procedure,

$$\mathbf{x}_0 = \arg\max_{\mathbf{x}} log\, p_\theta(\mathbf{y}_0|\mathbf{x}) = \arg\min_{\mathbf{x}} ||\mathbf{y}_0 - \mathcal{A}(\mathbf{x})||_2^2, \tag{3}$$

where $\mathbf{x}$ is initialized to $\mathcal{D}(\hat{\mathbf{z}}_0)$ for $\hat{\mathbf{z}}_0 \sim \mathcal{N}(0, \mathbf{I})$. Then, after reaching convergence, an initial $\mathbf{z}_0$ is obtained by applying the encoder on the outcome, namely $\mathbf{z}_0 = \mathcal{E}(\mathbf{x}_0)$. An alternative for this procedure is to perform the optimization process directly in the latent space. However, in our experiments we found that the former option worked better and was substantially faster as it did not involve expensive gradient propagation through the decoder.

### 4.2.2. POSTERIOR SAMPLING

We now move on to explain step 3(b) of the sampling procedure. Due to the non-linearity of the decoder, even for linear corruption operators $\mathcal{A}$, finding the exact posterior is intractable. One option to overcome this difficulty is to use SMC sampling. In what follows, we first describe approximate (tractable) target distributions for $\{p_\theta(\mathbf{z}_{t:T}|\mathbf{y}_{0:T})\}_{t=0}^{t=T}$, then we present the induced sequence weights $w_t$ and a novel proposal distributions $\pi_t$ for all timesteps $t$. These components are used as part of a general sequential importance sampling with resampling (SISR) algorithm as depicted in Algorithm 1.

**Approximate target distributions.** First, notice that due to the structure of the model, the posterior density of the r.v. $\mathbf{z}_t$ at each step $t$ depends only on $\mathbf{z}_{t+1:T}$. Hence, the target distribution of interest at each timestep $t$ is $p_\theta(\mathbf{z}_{t:T}|\mathbf{y}_{0:T})$. However, even computing the unnormalized quantity of that target distribution can be costly. Therefore, we make the following assumption $p_\theta(\mathbf{z}_{t:T}|\mathbf{y}_{0:T}) \approx p_\theta(\mathbf{z}_{t:T}|\mathbf{y}_{t:T}, \mathbf{y}_0)$. This assumption is reasonable since $\mathbf{y}_0$ stores all the input information to begin with. In Appendix B we arrive at the following recursive formula which forms the approximate target distributions of the SMC procedure for all $t$:

$$p_\theta(\mathbf{z}_{t:T}|\mathbf{y}_{t:T}, \mathbf{y}_0) \propto$$
$$\begin{cases} p_\theta(\mathbf{y}_T|\mathbf{z}_T)\bar{p}_\theta(\mathbf{y}_0|\mathbf{z}_T)p(\mathbf{z}_T), & t = T, \\ \frac{p_\theta(\mathbf{y}_t|\mathbf{z}_t)\bar{p}_\theta(\mathbf{y}_0|\mathbf{z}_t)}{\bar{p}_\theta(\mathbf{y}_0|\mathbf{z}_{t+1})}p_\theta(\mathbf{z}_t|\mathbf{z}_{t+1})p_\theta(\mathbf{z}_{t+1:T}|\mathbf{y}_{t+1:T}, \mathbf{y}_0), \\ & 0 < t < T, \\ \frac{p_\theta(\mathbf{y}_0|\mathbf{z}_0)}{\bar{p}_\theta(\mathbf{y}_0|\mathbf{z}_1)}p_\theta(\mathbf{z}_0|\mathbf{z}_1)p_\theta(\mathbf{z}_{1:T}|\mathbf{y}_{0:T}), & t = 0, \end{cases}$$
$$\tag{4}$$

where we define $\bar{p}_\theta(\mathbf{y}_0|\mathbf{z}_t) := p_\theta(\mathbf{y}_0|\bar{\mathbf{z}}_0(\mathbf{z}_t)) = \mathcal{N}(\mathbf{y}_0|\mathcal{A}(\mathcal{D}(\bar{\mathbf{z}}_0(\mathbf{z}_t))), (1-\bar{\alpha}_t)\mathbf{I}))$, with the variance taken to be the variance of $\mathbf{z}_t|\mathbf{z}_0$ according to the forward process. Importantly, as the final target distribution at $t = 0$ matches the desired distribution $p_\theta(\mathbf{z}_{0:T}|\mathbf{y}_{0:T})$ (see Eq. 10 in Appendix C), in the large compute limit samples from the correct target can be obtained despite all approximations.

**Sequence weights and proposal distributions.** To derive an SMC procedure using the proposed target distributions, it is common to use importance sampling. The

Table 1: Quantitative results on 1024 examples of size $256 \times 256$ from FFHQ test set. All methods were evaluated under the same experimental setup using LDM. Lower ($\downarrow$) is better in all metrics.

| | Inpainting (Box) | | | Inpainting (Free-Form) | | | Gaussian Deblurring | | | Super-Resolution ($8\times$) | | |
|---|---|---|---|---|---|---|---|---|---|---|---|---|
| | FID | NIQE | LPIPS | FID | NIQE | LPIPS | FID | NIQE | LPIPS | FID | NIQE | LPIPS |
| LDPS | 39.81 | 7.592 | 0.236 | 40.17 | 7.609 | 0.212 | 29.30 | 6.538 | 0.237 | **29.64** | **6.412** | 0.282 |
| LTDS | 39.57 | 7.602 | 0.236 | 39.78 | 7.578 | 0.212 | 30.23 | 6.553 | 0.238 | 30.45 | **6.412** | 0.284 |
| ReSample | 86.79 | 7.142 | 0.230 | 37.01 | **6.622** | **0.151** | 39.80 | 7.441 | 0.275 | 59.23 | 7.307 | 0.356 |
| PSLD | 39.68 | 6.544 | 0.246 | 36.26 | 6.835 | 0.216 | 36.31 | 6.802 | 0.341 | 40.33 | 6.803 | 0.347 |
| PFLD | 39.06 | **6.509** | 0.245 | 36.43 | 6.817 | 0.215 | 37.16 | 6.751 | 0.343 | 38.11 | 6.832 | 0.345 |
| LatentDAPS | 60.24 | 9.999 | 0.257 | 54.40 | 8.766 | 0.223 | 54.28 | 9.496 | 0.283 | 70.24 | 10.17 | 0.344 |
| LD-SMC (1 particle) | **33.37** | 7.032 | 0.212 | 33.67 | 7.034 | 0.194 | **29.19** | 6.575 | **0.232** | 30.02 | 6.426 | **0.277** |
| LD-SMC (5 particles) | 33.87 | 7.066 | **0.211** | **33.60** | 7.021 | 0.194 | 29.47 | **6.528** | 0.233 | 30.62 | 6.455 | 0.278 |

key idea is to construct proposal distributions $\pi_t(\mathbf{z}_{t:T})$, one for each timestep, from which it is easy to sample, and approximate $p(\mathbf{z}_{t:T}|\mathbf{y}_{0:T})$ with the empirical distribution $\sum_{i=1}^{N} w_t^{(i)} \delta_{\mathbf{z}_{t:T}^{(i)}}(\mathbf{z}_{t:T})$. Where, $\delta_{\mathbf{z}_{t:T}}$ is the Dirac measure and the weight $w_t$ (presented here for $t < T$) is defined as,

$$
\begin{aligned}
w_t &\propto \frac{\tilde{p}_\theta(\mathbf{z}_{t:T}|\mathbf{y}_{t:T}, \mathbf{y}_0)}{\pi_t(\mathbf{z}_{t:T})} \\
&= \frac{\tilde{p}_\theta(\mathbf{z}_t|\mathbf{z}_{t+1}, \mathbf{y}_t, \mathbf{y}_0)}{\pi_t(\mathbf{z}_t|\mathbf{z}_{t+1})} \frac{\tilde{p}_\theta(\mathbf{z}_{t+1:T}|\mathbf{y}_{t+1:T}, \mathbf{y}_0)}{\pi_{t+1}(\mathbf{z}_{t+1:T})} \\
&= \frac{\tilde{p}_\theta(\mathbf{z}_t|\mathbf{z}_{t+1}, \mathbf{y}_t, \mathbf{y}_0)}{\pi_t(\mathbf{z}_t|\mathbf{z}_{t+1})} w_{t+1}.
\end{aligned}
\tag{5}
$$

Here, $\tilde{p}_\theta(\cdot)$ is the unnormalized approximate target density. Plugging the unnormalized target densities from Eq. 4 in Eq. 5 for all timesteps $t$ results in,

$$
w_t \propto \begin{cases} \frac{p_\theta(\mathbf{y}_T|\mathbf{z}_T) \bar{p}_\theta(\mathbf{y}_0|\mathbf{z}_T) p_\theta(\mathbf{z}_T)}{\pi_T(\mathbf{z}_T)}, & t = T, \\ \frac{p_\theta(\mathbf{y}_t|\mathbf{z}_t) \bar{p}_\theta(\mathbf{y}_0|\mathbf{z}_t) p_\theta(\mathbf{z}_t|\mathbf{z}_{t+1})}{\bar{p}_\theta(\mathbf{y}_0|\mathbf{z}_{t+1}) \pi_t(\mathbf{z}_t|\mathbf{z}_{t+1})} w_{t+1}, & 0 < t < T, \\ \frac{p_\theta(\mathbf{y}_0|\mathbf{z}_0) p_\theta(\mathbf{z}_0|\mathbf{z}_1)}{\bar{p}_\theta(\mathbf{y}_0|\mathbf{z}_1) \pi_0(\mathbf{z}_0|\mathbf{z}_1)} w_1, & t = 0. \end{cases}
\tag{6}
$$

With these weights, we now turn to defining the proposal distributions $\pi_t$ for all timesteps $t$. The optimal choice in the sense of minimizing the variance of the weights is $\pi(\mathbf{z}_t|\mathbf{z}_{t:T}) = p_\theta(\mathbf{z}_t|\mathbf{z}_{t+1}, \mathbf{y}_{0:t})$ (Doucet et al., 2000; Särkkä, 2013). However, they cannot be obtained in closed form. Hence, we design alternative proposal distributions. The proposal distributions for timesteps $t < T$ are defined to be Gaussian $\pi_t(\mathbf{z}_t|\mathbf{z}_{t+1}) = \mathcal{N}(\mathbf{m}_t, \mathbf{S}_t)$ with parameters:

$$
\begin{aligned}
\mathbf{S}_t &= \tilde{\sigma}_{t+1}^2 \mathbf{I} \\
\mathbf{m}_t &= \boldsymbol{\mu}_\theta(\mathbf{z}_{t+1}, t+1) - (\gamma_t \nabla_{\mathbf{z}_{t+1}} \log \bar{p}_\theta(\mathbf{y}_0|\mathbf{z}_{t+1}) \\
&\quad + \lambda_t \nabla_{\boldsymbol{\mu}_\theta(\mathbf{z}_{t+1}, t+1)} \log q_\theta(\mathbf{y}_t|\mathbf{z}_{t+1})).
\end{aligned}
\tag{7}
$$

Where $q_\theta(\mathbf{y}_t|\mathbf{z}_{t+1}) = \mathcal{N}(\mathbf{y}_t|\mathcal{A}(\mathcal{D}(\boldsymbol{\mu}_\theta(\mathbf{z}_{t+1}, t+1))), \tau^2 \mathbf{I})$, and $\gamma_t, \lambda_t$ are finite scaling coefficients. In addition, $\pi_T(\mathbf{z}_T)$, the distribution of the proposal at time $T$, is set to the diffusion prior distribution, $\pi_T(\mathbf{z}_T) = \mathcal{N}(0, \mathbf{I})$.

In Eq. 7, we set the variance to be the variance of the prior diffusion model, namely $\tilde{\sigma}_{t+1} = \sigma_{t+1}$; however, other choices are also applicable. The idea behind the proposal mean is to correct the prior mean estimation by shifting it toward latents that agree more strongly with both $\mathbf{y}_t$ and $\mathbf{y}_0$. Specifically, the second term of the correction can be seen as making one gradient update step starting from the current prior mean, which serves as an estimate to $\mathbf{z}_t$.

The parameters $\gamma_t$ and $\lambda_t$ control the effect of the correction terms to the prior mean. In practice, it is challenging to control the trade-off between the two correction terms. Hence, we propose the following simple approach. During the sampling process, the first term only is used, that is, $\lambda_t = 0$, and then starting from some predefined timestep $s$, the second term is used as well. The intuition here is that during the initial sampling steps, the quality of the labels $\mathbf{y}_t$ may not be good. Therefore, we rely on the first term through the posterior mean estimator to capture the general semantics of the image. However, in later sampling stages the quality of the labels increases (see Figure 7 in the appendix), and the latter correction term can help capture fine details in the image. This design choice also relates to the three-stage phenomenon in the diffusion sampling process witnessed in the literature (Yu et al., 2023). Setting $\lambda_t = 0$ for all timesteps reduces the LD-SMC proposal update to that of TDS (Wu et al., 2023). Appendix D shows an instantiation of $\gamma_t$ and $\lambda_t$ used in this study.

Having defined the weights and the proposal distributions, we present LD-SMC concisely in Algorithm 1. The algorithm has two parts; the first part is the Gibbs sampling process and the second is the SMC sampling using SISR. In the algorithm the abbreviation "Multi" refers to Multinomial distribution and a resampling step is performed at each iteration, although it is not mandatory. Appendix C provides a proof that LD-SMC can achieve an arbitrarily accurate estimate of $p_\theta(\mathbf{z}_0|\mathbf{y}_0)$ under several conditions. Importantly, the estimate does not depend on any of the approximations made to derive our model. The following

Theorem summarizes that informally,

**Theorem 4.1.** *(informal) Let* $\mathbb{P}_N(\mathbf{z}_{0:T}) = \sum_{i=1}^{N} w_0^{(i)} \delta_{\mathbf{z}_{0:T}^{(i)}}(\mathbf{z}_{0:T})$ *be the discrete measure obtained by the function* **SISR** *in Algorithm 1. Under regularity conditions* $\mathbb{P}_N(\mathbf{z}_{0:T})$ *converges setwise to* $p_\theta(\mathbf{z}_{0:T}|\mathbf{y}_{0:T})$ *as* $N \to \infty$. *Furthermore, the stationary distribution of the Gibbs sampling process is* $p_\theta(\mathbf{z}_{0:T}, \mathbf{y}_{1:T}|\mathbf{y}_0)$, *and the marginal* $p_\theta(\mathbf{z}_0|\mathbf{y}_0)$ *is the limiting distribution of the* $\mathbf{z}_0$'s *subchain.*

**Connection to TDS.** While the derivation is different, the SMC procedure in LD-SMC resembles somewhat that of TDS (Wu et al., 2023). The main difference between the two methods is the dependence on the auxiliary variables $\mathbf{y}_{1:T}$, which are only relevant for LD-SMC in both the proposal distributions and the weights. Empirically, we observed that this additional conditioning helped to better align the sampling with the corrupted image $\mathbf{y}_0$ compared to using the posterior mean approximation as in (Chung et al., 2023b) and (Wu et al., 2023).

## 5. Experiments

### 5.1. Experimental Setting

We evaluated LD-SMC on ImageNet (Russakovsky et al., 2015) and FFHQ (Karras et al., 2019); both are common in the literature of inverse problems (Chung et al., 2023b; Dou & Song, 2024). In ImageNet, samples were conditioned on the class label. The guidance scale was fixed to $1.0$ in all our experiments. Results can be improved by adjusting it (Rombach et al., 2022). Images were resized to $3 \times 256 \times 256$ and normalized to the range $[0, 1]$. We used the latent diffusion models VQ-4 / CIN256-V2 (Rombach et al., 2022) as the prior model in FFHQ / ImageNet, respectively, with the DDIM diffusion sampler (Song et al., 2021a), according to the data split in (Esser et al., 2021). We sampled 1024 random images from the validation set of each dataset which were used to evaluate all methods. We followed the protocol of (Song et al., 2024) and added Gaussian noise with zero mean and standard deviation $\tau = 0.01$ to the corrupted images. Full experimental details are provided in Appendix A.

**Compared methods.** We compare LD-SMC with several recent SoTA baseline methods. All methods were evaluated in a similar experimental setup using the same latent diffusion model to ensure fairness in the comparisons. **(1) DPS** (Chung et al., 2023b), which introduces correction to the sampling process of the diffusion through the posterior mean estimator; **(2) TDS** (Wu et al., 2023), which uses the twisting technique for approximate sequential Monte Carlo sampling; since DPS and TDS were designed for pixel-space diffusion models, we term the adapted versions of them to the latent space LDPS and LTDS, respectively.

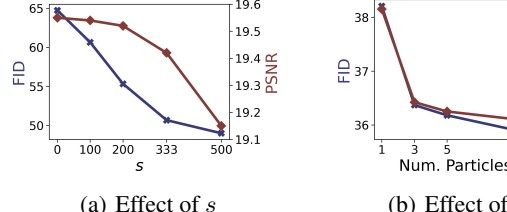

(a) Effect of $s$      (b) Effect of $N$

Figure 5: FID and PSNR values when varying $s$ (left) and the number of particles $N$ (right) on ImageNet box (left) and free-form (right) inpainting tasks.

**(3) ReSample** (Song et al., 2024), which applies an optimization procedure during the sampling process to match the approximate posterior mean to the label followed by a resampling step; **(4) PSLD** (Rout et al., 2023), which introduces a correction term to the DPS step to "glue" $\mathbf{z}_0$; **(5) PFLD** (Nazemi et al., 2024), a particle filtering approach for LDM based on PSLD update for the proposal distribution; **(6) LatentDAPS** which decouples the latents in the sampling trajectory. In all experiments, we evaluated LD-SMC with $N = \{1, 5\}$ particles. Using one particle incurs a computational cost similar to that of LDPS (see Table 4 in the appendix). LTDS and PFLD were evaluated with $N = 5$ particles. All methods were tested on the following inverse problem tasks: box inpainting, Gaussian deblurring, super-resolution ($8\times$) and free-form inpainting. We followed the setup in (Chung et al., 2023b) for the first three tasks and the setup in (Saharia et al., 2022) for the last one with a few changes to the default hyperparameters. In inpainting tasks, the masks were randomly sampled for each new sample.

**Evaluation metrics.** In the main text we report the following metrics, FID (Heusel et al., 2017), NIQE (Mittal et al., 2012), and LPIPS (Zhang et al., 2018). Full results including PSNR and SSIM (Wang et al., 2004) are deferred to Appendix H. FID and NIQE are considered perceptual metrics; lower values in them indicate higher perceptual quality. The other metrics can be considered as distortion metrics, which quantify some discrepancy between the generated images and the ground-truth values. Since perceptual metrics and distortion metrics can be in conflict with each other (Blau & Michaeli, 2018), we put more emphasis on perceptual quality. Hence, for all methods, we performed grid search over hyper-parameters and chose the best configuration according to the FID.

### 5.2. Experimental Results

Quantitative results are shown in Tables 1 and 2. From the tables, LD-SMC is usually the best or second best among all the comparisons. Specifically, on inpainting where extrapolation is needed within the masked region and details should

Table 2: Quantitative results on 1024 examples of size $256 \times 256$ from ImageNet test set. All methods were evaluated under the same experimental setup using LDM. Lower ($\downarrow$) is better in all metrics.

| | Inpainting (Box) | | | Inpainting (Free-Form) | | | Gaussian Deblurring | | | Super-Resolution ($8\times$) | | |
|---|---|---|---|---|---|---|---|---|---|---|---|---|
| | FID | NIQE | LPIPS | FID | NIQE | LPIPS | FID | NIQE | LPIPS | FID | NIQE | LPIPS |
| LDPS | 65.04 | 7.935 | 0.379 | 53.47 | 7.867 | 0.334 | 52.48 | 6.855 | 0.383 | 61.02 | 6.514 | 0.439 |
| LTDS | 64.74 | 7.907 | 0.378 | 52.75 | 7.884 | 0.334 | 50.82 | 6.695 | 0.379 | 59.12 | 6.270 | 0.435 |
| ReSample | 90.32 | 8.464 | **0.318** | 44.15 | 7.104 | **0.248** | **46.45** | 7.411 | **0.353** | 87.65 | 8.290 | 0.491 |
| PSLD | 71.15 | 8.042 | 0.434 | 62.38 | 8.037 | 0.411 | 60.68 | **6.599** | 0.417 | 66.56 | 7.669 | 0.489 |
| PFLD | 72.83 | 7.933 | 0.436 | 63.34 | 8.026 | 0.414 | 60.94 | 6.733 | 0.421 | 64.72 | 7.685 | 0.492 |
| LatentDAPS | 98.24 | 11.36 | 0.394 | 68.65 | 9.625 | 0.330 | 77.09 | 10.55 | 0.417 | 104.6 | 12.68 | 0.489 |
| LD-SMC (1 particle) | 51.49 | **6.878** | 0.326 | 38.21 | 6.969 | 0.285 | 52.48 | 6.855 | 0.383 | 58.06 | 6.243 | 0.434 |
| LD-SMC (5 particles) | **50.67** | 6.891 | 0.325 | **36.18** | **6.671** | 0.278 | 52.29 | 6.791 | 0.383 | 57.89 | **6.238** | **0.433** |

be preserved outside of it, LD-SMC can greatly improve over baseline methods. This property is also manifested in Figure 3 and Figure 8 in the appendix. LD-SMC manages to produce plausible reconstructions while maintaining fine details. The differences are especially highlighted on the ImageNet dataset which has more diversity in it. Also, as is clear from the figures, ReSample images suffer from significant artifacts. We speculate that it partly stems from the complete optimization process performed in every few sampling steps according to this method.

Figure 4 shows qualitative results for Gaussian deblurring of images by all methods. From the figure, LD-SMC, LDPS, LTDS, and ReSample are able to generate plausible reconstructions on this task. PSLD and PFLD images have noticeable artifacts, while LatantDAPS images are blurry.

Additional quantitative and qualitative results can be found in Appendix H and I. In terms of the distortion metrics PSNR and SSIM, LD-SMC usually substantially outperforms PSLD and PFLD and has a small advantage over LDPS and LTDS, but overall the three methods are comparable. ReSample and LatentDAPS are usually the best. Nevertheless, as can be seen visually and as indicated by other metrics, it comes at the expense of perceptual quality. LatentDAPS images are overly smooth and details are not preserved in them, while ReSample images have noticeable artifacts, especially in inpainting tasks.

### 5.3. Analysis

In Figure 5 we evaluate LD-SMC in terms of FID and PSNR on inpainting tasks when varying $s$, the hyperparameter that controls the influence of $\mathbf{y}_t$ in the proposal distribution, and $N$, the number of particles. From Figure 5(a) there is a clear trade-off in FID and PSNR when $s$ changes. Higher $s$ values result in a better FID at the expense of PSNR, while lower $s$ values show the opposite trend. For inpainting, $s = 333$ strikes a good balance between the two metrics while giving more emphasis to the perceptual quality. In addition, note that taking $s = 0$ is similar to using the DPS as a proposal

distribution which results in a significant reduction in perceptual quality. Figure 5(b) shows an improvement in FID while suffering from a small reduction in PSNR when increasing the number of particles. In our comparisons, we chose $N = 5$ to balance between performance and computational cost incurred by adding particles (see Section E for further details). Empirically, we found that sometimes only increasing the number of particles resulted in favorable metric values, as is also clear from the tables. We attribute this to the relatively small number of particles with which we experimented (up to 10). However, in general, one can expect that as the number of particles increases, the samples will become more accurate. Finally, in Appendix F we present an ablation study on the effect of using several Gibbs steps. We show that improvements in FID and PSNR can be obtained by taking more than one step, yet there is not always a clear trend of improvements over the steps.

## 6. Conclusion

In this study, we presented LD-SMC, a novel method for solving inverse problems in the latent space of diffusion models using SMC. LD-SMC augment the model with auxiliary observations, one per each timestep, and use these observations to guide the sampling process as part of the reverse diffusion process. This framework can be seen as applying one step of blocked Gibbs sampling. To perform SMC sampling, we suggested a novel weighing scheme and proposal distributions. Both are based on information from the auxiliary labels and the true label $\mathbf{y}_0$. Empirically, we validated LD-SMC against strong baseline methods on common benchmarks. The results suggest that LD-SMC can improve the performance over baseline methods, especially in cases where extrapolation is needed (e.g., in inpainting tasks). one limitation of our approach is related to computational demand. The sampling time and the memory demand increase with the number of particles and Gibbs iterations. This effect can be partially mitigated by taking fewer particles and GPU parallelization.

## Impact Statement

This paper presents work whose goal is to advance the field of Machine Learning. There are many potential societal consequences of our work, none which we feel must be specifically highlighted here.

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

# Appendix of Inverse Problem Sampling in Latent Space Using Sequential Monte Carlo

## A. Full Experimental Details

The experiments were carried out mainly using an NVIDIA A100 having 40GB and 80GB memory. In all experiments, we used the DDIM formulation (Song et al., 2021a), although LD-SMC can be applied with other sampling procedures. For all methods, we performed a hyperparameter search on $\eta \in \{0.05, 0.5, 1.0\}$ and found that LD-SMC worked best with $\eta = 1.0$. For DPS and TDS we examined several scaling coefficient schemes for the prior mean update, including the ones proposed in each of the corresponding papers, and found that our proposed update worked better for both. For all three methods (LD-SMC, DPS, and TDS) we searched for $\kappa_1 \in \{0.3, 0.4, 0.5, 1.0, 1.5, 2.0\}$. For our method, we also performed a grid search over $\kappa_2 \in \{0.5, 1.5, 2.5\}$, $s \in \{0, 100, 200, 333\}$, and $\rho \in \{0.5, 0.75\}$. In addition, when applying resampling, we set the variance of $p_\theta(\mathbf{y}_t|\mathbf{z}_t)$ to $(1 - \bar{\alpha}_t)\mathbf{I}$ to match the variance of $\bar{p}_\theta(\mathbf{y}_0|\mathbf{z}_t)$. For PSLD, in most cases, the default hyperparameters suggested in the paper and code did not yield good results. Hence, we performed a grid search over PSLD's hyperparameters $\gamma_t \in \{1e-4, 1e-3, 1e-2, 5e-2, 0.1, 0.2\}$ and $\eta_t \in \{0.05, 0.1, 0.2, 0.5, 0.9\}$, and used the best ones for both PSLD and PFLD. For ReSample, we found that using $\eta = 0.0$, the default value in the code, usually performed best. Also, we performed a grid search over $\gamma$, the scaling coefficient of the resampling step std in $\{4, 8, 16, 40, 80, 200, 400\}$. For LatentDAPS, we used the default hyperparameters suggested in the paper, except for super-resolution tasks where we made a grid search over $\eta_0 \in \{1e-4, 5e-5, 1e-5, 5e-6, 1e-6, 5e-7, 1e-7\}$. In addition, for comparability with other methods, we used observation noise of $0.01$, instead of the default $0.05$ in the code, and in ImageNet experiments we used $N = 100$ steps. To choose the best set of hyperparameters, we evaluated each method both visually and using the FID on a sample of $64$ images. Then, we sampled $1024$ images using the best configuration. Similarly to ReSample, we found that applying an optimization process at the end of the sampling process in the latent space can sometimes improve visibility and metric values. We evaluated all models except LatentDAPS with and without this final optimization process and chose the best according to the FID. The optimization procedure was not applied to inpainting tasks since it created non-smooth changes at the boundaries of the box, making the images look non-natural.

For LD-SMC, we present here the chosen hyperparameters:

Table 3: LD-SMC hyperparameters for all tasks.

| Dataset | Task | $\kappa_1$ | $\kappa_2$ | $s$ | $\rho$ | Final Latent Optimization |
|---------|------|-----|-----|-----|-----|---------------------------|
| FFHQ | Inpainting (Box) | 1.0 | 2.5 | 333 | 0.75 | No |
| | Inpainting (Free-Form) | 1.0 | 2.5 | 333 | 0.75 | No |
| | Gaussian deblurring | 1.5 | 1.5 | 100 | 0.5 | Yes |
| | Super-resolution | 1.0 | 2.5 | 100 | 0.5 | Yes |
| ImageNet | Inpainting (Box) | 2.0 | 2.5 | 333 | 0.75 | No |
| | Inpainting (Free-Form) | 2.0 | 2.5 | 333 | 0.75 | No |
| | Gaussian deblurring | 0.5 | – | 0 | – | Yes |
| | Super-resolution | 0.3 | 2.5 | 333 | 0.5 | No |

## B. Approximate Target Distributions

In Section 4.2.2 we presented approximate target distributions for the SMC sampling procedure. Here we present the derivation for each case, namely (1) $t = T$, (2) $0 < t < T$, and (3) $t = 0$:

(1) $\underline{t = T}$ :

$$
\begin{aligned}
p_\theta(\mathbf{z}_T|\mathbf{y}_T,\mathbf{y}_0) &\propto p_\theta(\mathbf{y}_T|\mathbf{z}_T,\mathbf{y}_0)p_\theta(\mathbf{z}_T|\mathbf{y}_0) \\
&\propto p_\theta(\mathbf{y}_T|\mathbf{z}_T)p_\theta(\mathbf{y}_0|\mathbf{z}_T)p(\mathbf{z}_T) \\
&\approx p_\theta(\mathbf{y}_T|\mathbf{z}_T)\bar{p}_\theta(\mathbf{y}_0|\mathbf{z}_T)p(\mathbf{z}_T).
\end{aligned}
$$

Where, in the first transition we used Bayes rule, in the second transition we used the model dependencies and Bayes rule again, and in the last transition we make the additional approximation of conditioning on the posterior mean estimator.

(2) $\underline{0 < t < T}$ :

$$
\begin{aligned}
p_\theta(\mathbf{z}_{t:T}|\mathbf{y}_{t:T},\mathbf{y}_0) &\propto p_\theta(\mathbf{y}_t|\mathbf{z}_{t:T},\mathbf{y}_{t+1:T},\mathbf{y}_0)p_\theta(\mathbf{z}_{t:T}|\mathbf{y}_{t+1:T},\mathbf{y}_0) \\
&= p_\theta(\mathbf{y}_t|\mathbf{z}_t)p_\theta(\mathbf{z}_t|\mathbf{z}_{t+1:T},\mathbf{y}_{t+1:T},\mathbf{y}_0)p_\theta(\mathbf{z}_{t+1:T}|\mathbf{y}_{t+1:T},\mathbf{y}_0) \\
&= p_\theta(\mathbf{y}_t|\mathbf{z}_t)p_\theta(\mathbf{z}_t|\mathbf{z}_{t+1},\mathbf{y}_0)p_\theta(\mathbf{z}_{t+1:T}|\mathbf{y}_{t+1:T},\mathbf{y}_0) \\
&= p_\theta(\mathbf{y}_t|\mathbf{z}_t)\frac{p_\theta(\mathbf{y}_0|\mathbf{z}_t)}{p_\theta(\mathbf{y}_0|\mathbf{z}_{t+1})}p_\theta(\mathbf{z}_t|\mathbf{z}_{t+1})p_\theta(\mathbf{z}_{t+1:T}|\mathbf{y}_{t+1:T},\mathbf{y}_0) \\
&\approx \frac{p_\theta(\mathbf{y}_t|\mathbf{z}_t)\bar{p}_\theta(\mathbf{y}_0|\mathbf{z}_t)}{\bar{p}_\theta(\mathbf{y}_0|\mathbf{z}_{t+1})}p_\theta(\mathbf{z}_t|\mathbf{z}_{t+1})p_\theta(\mathbf{z}_{t+1:T}|\mathbf{y}_{t+1:T},\mathbf{y}_0).
\end{aligned}
$$

Where, in the first transition we used Bayes rule, in the third transition we used the Markovian assumption, in the forth transition we used Bayes rule again, and in the last transition we make an additional approximation and condition on the posterior mean estimator for both time $t$ and time $t + 1$.

(3) $\underline{t = 0}$ :

$$
\begin{aligned}
p_\theta(\mathbf{z}_{0:T}|\mathbf{y}_{1:T},\mathbf{y}_0) &= p_\theta(\mathbf{z}_0|\mathbf{z}_{1:T},\mathbf{y}_{0:T})p_\theta(\mathbf{z}_{1:T}|\mathbf{y}_{1:T},\mathbf{y}_0) \\
&= p_\theta(\mathbf{z}_0|\mathbf{z}_1,\mathbf{y}_0)p_\theta(\mathbf{z}_{1:T}|\mathbf{y}_{0:T}) \\
&= \frac{p_\theta(\mathbf{y}_0|\mathbf{z}_0)}{p_\theta(\mathbf{y}_0|\mathbf{z}_1)}p_\theta(\mathbf{z}_0|\mathbf{z}_1)p_\theta(\mathbf{z}_{1:T}|\mathbf{y}_{0:T}) \\
&\approx \frac{p_\theta(\mathbf{y}_0|\mathbf{z}_0)}{\bar{p}_\theta(\mathbf{y}_0|\mathbf{z}_1)}p_\theta(\mathbf{z}_0|\mathbf{z}_1)p_\theta(\mathbf{z}_{1:T}|\mathbf{y}_{0:T})
\end{aligned}
$$

Where, in the third transition we used Bayes rule, and in the last transition we make an additional approximation and condition on the posterior mean estimator.

## C. Asymptotic Accuracy of LD-SMC

Here we provide a proof that $\mathbf{z}_0$ samples can be made arbitrarily accurate according to our model design. Our proof is composed of three parts, all of which have been previously established in the literature. We will restate them here and accommodate them to our setting. The three parts are, (1) augmenting the model with auxiliary random variables, ergo performing a completion of the desired marginal density $p_\theta(\mathbf{z}_0|\mathbf{y}_0)$ (Definition 10.3 in (Robert & Casella, 1999)), (2) Proving asymptotic accuracy of the SMC procedure following a similar lines of Theorem 2 in (Wu et al., 2023), (3) Showing sufficient conditions that the Markov chain generated by the Gibbs sampling procedure is ergodic and hence $p_\theta(\mathbf{z}_0|\mathbf{y}_0)$ is the limiting distribution from which $\mathbf{z}_0$'s are sampled (Theorem 10.6 in (Robert & Casella, 1999)).

First of all, here we provide a concise version of our Gibbs sampler presented in Section 4.2. We denote by $j$ the index of the Gibbs iterations:

1. Initialize $\mathbf{z}_{0:T}^0$

2. For $j = 0, ..., J - 1$:
    (a) Sample, $\mathbf{y}_{1:T}^{j+1} \sim p_\theta(\mathbf{y}_{1:T}|\mathbf{z}_{0:T}^j,\mathbf{y}_0)$.
    (b) Sample, $\mathbf{z}_{0:T}^{j+1} \sim p_\theta(\mathbf{z}_{0:T}|\mathbf{y}_{1:T}^{j+1},\mathbf{y}_0)$ using SMC.

**Definition C.1.** (Robert & Casella, 1999). Given a probability density $p_\theta(\mathbf{z}_0|\mathbf{y}_0)$, a density $g$ that satisfies $\int g(\mathbf{z}_0,\mathbf{z}_{1:T},\mathbf{y}_{1:T}|\mathbf{y}_0)d\mathbf{z}_{1:T}d\mathbf{y}_{1:T}$ is called a completion of $p_\theta(\mathbf{z}_0|\mathbf{y}_0)$.

We denote by $(\mathbf{z}_{0:T}^j,\mathbf{y}_{1:T}^j)$ the Markov chain generated by our proposed Gibbs sampler. Similarly, denote by $(\mathbf{z}_0^j)$ the corresponding subchain.

**Theorem C.2.** *(Robert & Casella, 1999). For the Gibbs sampler described in Section 4.2, if $(\mathbf{z}_{0:T}^j, \mathbf{y}_{1:T}^j)$ is ergodic, then the distribution $g$ is a stationary distribution for it and $p_\theta(\mathbf{z}_0|\mathbf{y}_0)$ is the limiting distribution of the subchain $(\mathbf{z}_0^j)$.*

*Proof.* The support of $g$ is $\mathbb{R}^d$, with $d = dim(\mathbf{z}_{0:T}, \mathbf{y}_{1:T})$, and hence is connected. Also, since each conditional distribution in the Gibbs sampling process is a Gaussian or a multiplication of Gaussian densities, they are all strictly positive. Following Lemma 10.11 in (Robert & Casella, 1999), $(\mathbf{z}_{0:T}^j, \mathbf{y}_{1:T}^j)$ is irreducible and aperiodic, i.e., it is ergodic. $\square$

Specifically, Theorem C.2 shows that in order to obtain samples from the marginal distribution $p_\theta(\mathbf{z}_0|\mathbf{y}_0)$, we need to sample $\mathbf{y}_{1:T}$ and $\mathbf{z}_{0:T}$ from the conditional distribution according to the model at each iteration. Sampling $\mathbf{y}_{1:T}$ variables is straightforward as it requires sampling from independent Gaussian distributions, that is, $p_\theta(\mathbf{y}_t|\mathbf{z}_t) = \mathcal{N}(\mathbf{y}_t|\mathcal{A}(\mathcal{D}(\mathbf{z}_t)), \tau^2\mathbf{I})$ for all $t > 0$. On the other hand, sampling $\mathbf{z}_{0:T}$ requires sampling using an SMC procedure. As we show next in the large particle limit, samples from $p_\theta(\mathbf{z}_{0:T}|\mathbf{y}_{0:T})$ are accurate.

To prove the asymptotic accuracy of the SMC procedure in LD-SMC, we adopt the formulation of (Wu et al., 2023). In what follows, to prevent cluttered notation, we drop the index of the Gibbs iteration and note that all quantities are those of a specific iteration. Specifically, here we first reiterate the following 3 important quantities, the prior distribution $p_\theta(\mathbf{z}_{0:T})$, the proposal distributions $\pi_t(\mathbf{z}_t|\mathbf{z}_{t+1})$, and the weighting functions $w_t(\mathbf{z}_t, \mathbf{z}_{t+1})$ and then proceed to the proof.

**Prior distribution**. Let $p_\theta(\mathbf{z}_{0:T})$ denote the diffusion generative model defined according to Eq. 1. Then, the Markovian structure of the prior diffusion model goes as follows,

$$\begin{cases} p_\theta(\mathbf{z}_t|\mathbf{z}_{t+1}) = \mathcal{N}(\boldsymbol{\mu}_\theta(\mathbf{z}_{t+1}, t+1), \sigma_{t+1}^2\mathbf{I}) \\ \qquad = \mathcal{N}\left(\mathbf{z}_t|\sqrt{\bar{\alpha}_t}\left(\frac{\mathbf{z}_{t+1}-\sqrt{1-\bar{\alpha}_{t+1}}\cdot\boldsymbol{\epsilon}_\theta(\mathbf{z}_{t+1},t+1)}{\sqrt{\bar{\alpha}_{t+1}}}\right) + \sqrt{1-\bar{\alpha}_t-\sigma_{t+1}^2}\cdot\boldsymbol{\epsilon}_\theta(\mathbf{z}_{t+1},t+1), \sigma_{t+1}^2\mathbf{I}\right), & 1 < t < T, \\ p_\theta(\mathbf{z}_T) = \mathcal{N}(0, \mathbf{I}) & t = T. \end{cases}$$

**Proposal distributions**. Denote the proposal distribution for timestep $t < T$ as $\pi_t(\mathbf{z}_t|\mathbf{z}_{t+1}) = \mathcal{N}(\mathbf{m}_t, \mathbf{S}_t)$, with parameters

$$\begin{aligned} \mathbf{S}_t &= \tilde{\sigma}_{t+1}^2\mathbf{I} \\ \mathbf{m}_t &= \boldsymbol{\mu}_\theta(\mathbf{z}_{t+1}, t+1) - (\gamma_t\nabla_{\mathbf{z}_{t+1}}\log\bar{p}_\theta(\mathbf{y}_0|\mathbf{z}_{t+1}) + \lambda_t\nabla_{\boldsymbol{\mu}_\theta(\mathbf{z}_{t+1},t+1)}\log q_\theta(\mathbf{y}_t|\mathbf{z}_{t+1})). \end{aligned} \tag{8}$$

Where $\bar{p}_\theta(\mathbf{y}_0|\mathbf{z}_{t+1}) = \mathcal{N}(\mathbf{y}_0|\mathcal{A}(\mathcal{D}(\bar{\mathbf{z}}_0(\mathbf{z}_{t+1}))), (1-\bar{\alpha}_{t+1})\mathbf{I})$, and $q_\theta(\mathbf{y}_t|\mathbf{z}_{t+1}) = \mathcal{N}(\mathbf{y}_t|\mathcal{A}(\mathcal{D}(\boldsymbol{\mu}_\theta(\mathbf{z}_{t+1}, t+1))), \tau^2\mathbf{I})$, and $\gamma_t, \lambda_t$ are finite scaling coefficients. The distribution of the proposal at time $T$, $\pi_T(\mathbf{z}_T)$, from which the initial sample is taken, is set to the diffusion prior distribution $p_\theta(\mathbf{z}_T) = \mathcal{N}(0, \mathbf{I})$.

**Weighting functions**. The unnormalized weighting functions for all timesteps $t$ are summarized as follows,

$$\begin{cases} \tilde{w}_T(\mathbf{z}_T) = p_\theta(\mathbf{y}_T|\mathbf{z}_T)\bar{p}_\theta(\mathbf{y}_0|\mathbf{z}_T), \\ \tilde{w}_t(\mathbf{z}_t, \mathbf{z}_{t+1}) = p_\theta(\mathbf{y}_t|\mathbf{z}_t)\bar{p}_\theta(\mathbf{y}_0|\mathbf{z}_t)p_\theta(\mathbf{z}_t|\mathbf{z}_{t+1})/(\bar{p}_\theta(\mathbf{y}_0|\mathbf{z}_{t+1})\pi_t(\mathbf{z}_t|\mathbf{z}_{t+1})), \\ \tilde{w}_0(\mathbf{z}_0, \mathbf{z}_1) = p_\theta(\mathbf{y}_0|\mathbf{z}_0)p_\theta(\mathbf{z}_0|\mathbf{z}_1)/\bar{p}_\theta(\mathbf{y}_0|\mathbf{z}_1)\pi_0(\mathbf{z}_0|\mathbf{z}_1), \end{cases}$$

where $p_\theta(\mathbf{y}_t|\mathbf{z}_t) = \mathcal{N}(\mathbf{y}_t|\mathcal{A}(\mathcal{D}(\mathbf{z}_t)), \tau^2\mathbf{I})$.

In SMC the proposal distributions and weighting functions define a sequence of intermediate target distributions $\{\nu_t\}_{t=0}^T$ defined as follows,

$$\nu_t(\mathbf{z}_{t:T}) = \frac{1}{\mathcal{L}_t}\left(\pi_T(\mathbf{z}_T)\prod_{t'=t}^T\pi_{t'}(\mathbf{z}_{t'}|\mathbf{z}_{t'+1})\right)\left(\tilde{w}_T(\mathbf{z}_T)\prod_{t'=t}^T\tilde{w}_{t'}(\mathbf{z}_{t'}, \mathbf{z}_{t'+1})\right). \tag{9}$$

Importantly, if for given weighting functions and proposal distributions the final target distribution $\nu_0$ coincides with the desired posterior distribution $p(\mathbf{z}_{0:T}|\mathbf{y}_{0:T})$, then samples that are approximately distributed according to $p(\mathbf{z}_{0:T}|\mathbf{y}_{0:T})$ can be obtained (Doucet et al., 2001a; Naesseth et al., 2019). The approximation becomes accurate in the limit of large number of particles. For $t = 0$, plugging in Eq. 9 the proposal distributions and weighting functions and observing that all appearances

of the proposal distributions besides that of time $T$ cancel out, we obtain:

$$\nu_0(\mathbf{z}_{0:T}) = \frac{1}{\mathcal{L}_0} p_\theta(\mathbf{z}_T) p_\theta(\mathbf{y}_T|\mathbf{z}_T) \bar{p}_\theta(\mathbf{y}_0|\mathbf{z}_T) \left[ \prod_{t=1}^{T-1} \frac{p_\theta(\mathbf{y}_t|\mathbf{z}_t) \bar{p}_\theta(\mathbf{y}_0|\mathbf{z}_t) p_\theta(\mathbf{z}_t|\mathbf{z}_{t+1})}{\bar{p}_\theta(\mathbf{y}_0|\mathbf{z}_{t+1})} \right] \frac{p_\theta(\mathbf{y}_0|\mathbf{z}_0)}{\bar{p}_\theta(\mathbf{y}_0|\mathbf{z}_1)} p_\theta(\mathbf{z}_0|\mathbf{z}_1)$$

$$= \frac{1}{\mathcal{L}_0} \prod_{t=0}^{T} p_\theta(\mathbf{y}_t|\mathbf{z}_t) \prod_{t=0}^{T-1} p_\theta(\mathbf{z}_t|\mathbf{z}_{t+1}) p_\theta(\mathbf{z}_T) = \frac{1}{\mathcal{L}_0} p_\theta(\mathbf{y}_{0:T}|\mathbf{z}_{0:T}) p_\theta(\mathbf{z}_{0:T}) = p_\theta(\mathbf{z}_{0:T}|\mathbf{y}_{0:T}). \tag{10}$$

Where in the second equality $\bar{p}_\theta$ terms cancel out and in the last equality we applied Bayes rule.

Now that we established necessary conditions to sample from $p_\theta(\mathbf{z}_{0:T}|\mathbf{y}_{0:T})$, the following theorem from (Chopin et al., 2020) characterizes the conditions under which SMC algorithms converge. We adopt the formulation of (Wu et al., 2023) and present it here, adapted to our case.

**Theorem C.3.** *((Chopin et al., 2020) – Proposition 11.4). Let $\{\mathbf{z}_{0:T}^{(i)}, w_0^{(i)}\}$ be the sequence of particles and final weights returned by the last iteration of the SMC algorithm with $N$ particles and using multinomial resampling. If the weight functions of all timesteps $w_t^{(i)}$ are positive and bounded, then for every $\nu_0$-measurable function $\phi$ of $\mathbf{z}_{0:T}^{(i)}$,*

$$\lim_{N\to\infty} \sum_{i=1}^{N} w_0^{(i)} \phi(\mathbf{z}_{0:T}^{(i)}) = \int \phi(\mathbf{z}_{0:T}) \nu_0(\mathbf{z}_{0:T}) d\mathbf{z}_{0:T}$$

*with probability one.*

Specifically, taking $\phi(\mathbf{z}_{0:T}) = \mathbb{I}[\mathbf{z}_{0:T} \in E]$ for any $\nu_0$-measurable set E implies the convergence of $\mathbb{P}_N(E) = \sum_{i=1}^{N} w_0^{(i)} \delta_{\mathbf{z}_{0:T}^{(i)}}(E)$. Here, $\delta_{\mathbf{z}}$ is the Dirac measure defined for a given point $\mathbf{z}$ and a $\nu_0$-measurable set E. The following proposition characterizes the conditions under which Theorem C.3 applies in our case.

**Proposition C.4.** *Let $\mathbb{P}_N(E) = \sum_{i=1}^{N} w_0^{(i)} \delta_{\mathbf{z}_{0:T}^{(i)}}(E)$ be a discrete distribution over particles with $\{(\mathbf{z}_{0:T}^{(i)}, w_0^{(i)})\}_{i=1}^{N}$ returned by the SMC procedure in Section 4.2.2 with $N$ particles. Assume for all $t$:*

*(a) The likelihood functions $p_\theta(\mathbf{y}_t|\mathbf{z}_t) = \mathcal{N}(\mathbf{y}_t|\mathcal{A}(\mathcal{D}(\mathbf{z}_t)), \tau^2\mathbf{I})$, the ratios $\bar{p}_\theta(\mathbf{y}_0|\mathbf{z}_t)/\bar{p}_\theta(\mathbf{y}_0|\mathbf{z}_{t+1})$, and $\bar{p}_\theta(\mathbf{y}_0|\mathbf{z}_T)$, are all positive and bounded.*

*(b) $\log \bar{p}_\theta(\mathbf{y}_0|\mathbf{z}_t)$ is continuous and has bounded gradients in $\mathbf{z}_t$.*

*(c) $\log q_\theta(\mathbf{y}_t|\mathbf{z}_{t+1})$ is continuous and has bounded gradients in $\boldsymbol{\mu}_\theta(\mathbf{z}_{t+1}, t+1)$.*

*(d) The proposal variance is larger than the prior diffusion model variance, namely $\tilde{\sigma}_t^2 > \sigma_t^2$.*

*Then $\mathbb{P}_N$ converges setwise to $p_\theta(\mathbf{z}_{0:T}|\mathbf{y}_{0:T})$ with probability one, that is for every set E, $\lim_{N\to\infty} \mathbb{P}_N(E) = \int_E p_\theta(\mathbf{z}_{0:T}|\mathbf{y}_{0:T}) d\mathbf{z}_{0:T}$.*

Note that here, unlike (Wu et al., 2023), we do not need to assume that $\bar{p}_\theta(\mathbf{y}_0|\mathbf{z}_0) = p_\theta(\mathbf{y}_0|\mathbf{z}_0)$ due to our revised posterior distribution. Here, however, we add assumption (c). Assumptions (b) and (c) are the strongest assumptions as they may not apply even for linear transformations. But, for sufficiently smooth decoder and diffusion model in the input ($\mathbf{z}_t$ or $\boldsymbol{\mu}_\theta(\mathbf{z}_{t+1}, t+1)$) with uniformly bounded gradients, the assumptions will hold.

*Proof.* To prove the statement, we need to show that (1) the target distribution of the SMC procedure at time $t = 0$ is $p_\theta(\mathbf{z}_{0:T}|\mathbf{y}_{0:T})$, and (2) the weighting functions $w_t(\mathbf{z}_t, \mathbf{z}_{t+1})$ of all timesteps are all positive and bounded. The first condition was already established in Eq. 10, hence we proceed to the second condition.

Since all the weights are defined through multiplications of density functions, they are strictly positive. To show that they are bounded, we first note that, by assumption (a), $w_T(\mathbf{z}_T)$ is bounded. To show that the intermediate weights are bounded it is enough to consider the log transformation of the unnormalized weights,

$$\log \tilde{w}_t(\mathbf{z}_t, \mathbf{z}_{t+1}) = \begin{cases} \log p_\theta(\mathbf{y}_t|\mathbf{z}_t) + \log \frac{\bar{p}_\theta(\mathbf{y}_0|\mathbf{z}_t)}{\bar{p}_\theta(\mathbf{y}_0|\mathbf{z}_{t+1})} + \log \frac{p_\theta(\mathbf{z}_t|\mathbf{z}_{t+1})}{\pi_t(\mathbf{z}_t|\mathbf{z}_{t+1})}, & 0 < t < T, \\ \log p_\theta(\mathbf{y}_0|\mathbf{z}_0) + \log \frac{p_\theta(\mathbf{z}_0|\mathbf{z}_1)}{\pi_0(\mathbf{z}_0|\mathbf{z}_1)}, & t = 0. \end{cases}$$

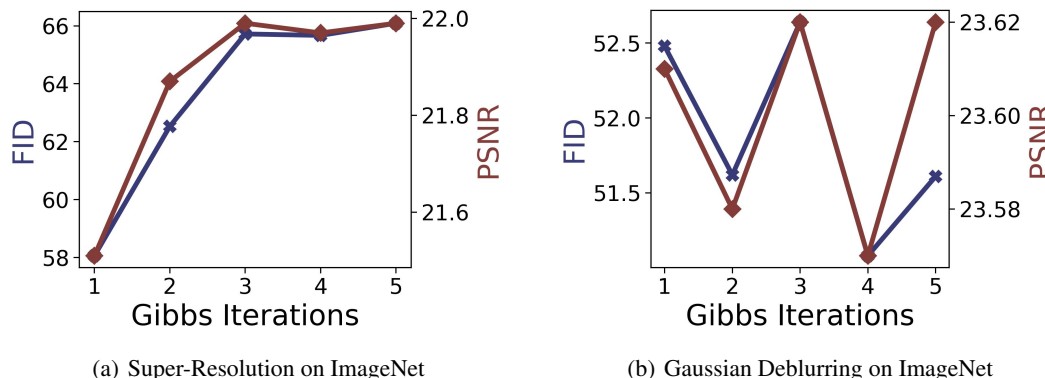

(a) Super-Resolution on ImageNet        (b) Gaussian Deblurring on ImageNet

Figure 6: FID and PSNR values when increasing the number of Gibbs iterations (with $N = 1$ particle).

By assumption (a), $\log p_\theta(\mathbf{y}_t|\mathbf{z}_t)$ and $\log \bar{p}_\theta(\mathbf{y}_0|\mathbf{z}_t)/\bar{p}_\theta(\mathbf{y}_0|\mathbf{z}_{t+1})$ are bounded for all $t$. We will show next that the $\log p_\theta(\mathbf{z}_t|\mathbf{z}_{t+1})/\pi_t(\mathbf{z}_t|\mathbf{z}_{t+1})$ terms are all bounded based on assumptions (b) - (d). We denote by $\boldsymbol{\mu}_t := \boldsymbol{\mu}_\theta(\mathbf{z}_{t+1}, t+1)$,

$$\log \frac{p_\theta(\mathbf{z}_t|\mathbf{z}_{t+1})}{\pi_t(\mathbf{z}_t|\mathbf{z}_{t+1})} = \log \frac{|2\pi\sigma_{t+1}^2\mathbf{I}|^{-0.5}\exp\{-(2\sigma_{t+1}^2)^{-1}||\mathbf{z}_t - \boldsymbol{\mu}_t||_2^2\}}{|2\pi\tilde{\sigma}_{t+1}^2\mathbf{I}|^{-0.5}\exp\{-(2\tilde{\sigma}_{t+1}^2)^{-1}||\mathbf{z}_t - \mathbf{m}_t||_2^2\}}$$

$$\stackrel{\text{c}}{=} -\frac{1}{2}[\sigma_{t+1}^{-2}||\mathbf{z}_t - \boldsymbol{\mu}_t||_2^2 - \tilde{\sigma}_{t+1}^{-2}||\mathbf{z}_t - \mathbf{m}_t||_2^2]$$

Expanding, $||\mathbf{z}_t - \mathbf{m}_t \pm \boldsymbol{\mu}_t||_2^2 = ||\mathbf{z}_t - \boldsymbol{\mu}_t||_2^2 + 2\langle\mathbf{z}_t - \boldsymbol{\mu}_t, \boldsymbol{\mu}_t - \mathbf{m}_t\rangle + ||\boldsymbol{\mu}_t - \mathbf{m}_t||_2^2$

$$= -\frac{1}{2}[(\sigma_{t+1}^{-2} - \tilde{\sigma}_{t+1}^{-2})||\mathbf{z}_t - \boldsymbol{\mu}_t||_2^2 - 2\tilde{\sigma}_{t+1}^{-2}\langle\mathbf{z}_t - \boldsymbol{\mu}_t, \boldsymbol{\mu}_t - \mathbf{m}_t\rangle - \tilde{\sigma}_{t+1}^{-2}||\boldsymbol{\mu}_t - \mathbf{m}_t||_2^2]$$

Notice, $||\boldsymbol{\mu}_t - \mathbf{m}_t||_2^2 = ||\gamma_t\nabla_{\mathbf{z}_{t+1}}\log\bar{p}_\theta(\mathbf{y}_0|\mathbf{z}_{t+1}) + \lambda_t\nabla_{\boldsymbol{\mu}_t}\log q_\theta(\mathbf{y}_t|\mathbf{z}_{t+1})||_2^2 < \infty$
by applying Minkowski inequality and using assumptions (b) and (c)

$$\stackrel{\text{c}}{=} -\frac{1}{2}[(\sigma_{t+1}^{-2} - \tilde{\sigma}_{t+1}^{-2})||\mathbf{z}_t - \boldsymbol{\mu}_t||_2^2 - 2\tilde{\sigma}_{t+1}^{-2}\langle\mathbf{z}_t - \boldsymbol{\mu}_t, \boldsymbol{\mu}_t - \mathbf{m}_t\rangle]$$

Apply Cauchy-Schwartz inequality,

$$\leq -\frac{1}{2}(\sigma_{t+1}^{-2} - \tilde{\sigma}_{t+1}^{-2})||\mathbf{z}_t - \boldsymbol{\mu}_t||_2^2 + \tilde{\sigma}_{t+1}^{-2}||\boldsymbol{\mu}_t - \mathbf{m}_t||_2||\mathbf{z}_t - \boldsymbol{\mu}_t||_2]$$

Apply the inequality $-\frac{a}{2}x^2 + bx \leq \frac{b^2}{2a}$ for $a = \sigma_{t+1}^{-2} - \tilde{\sigma}_{t+1}^{-2} > 0$ by assumption (d)

$$\leq \frac{\tilde{\sigma}_{t+1}^{-4}||\boldsymbol{\mu}_t - \mathbf{m}_t||_2^2}{2(\sigma_{t+1}^{-2} - \tilde{\sigma}_{t+1}^{-2})} < \infty.$$

Where the last inequality is again due to assumptions (b) and (c). Here, $\stackrel{\text{c}}{=}$ denotes an equality up to a constant. □

## D. Proposal Distribution Scaling Coefficients

Recall that our proposal distribution (Eq. 7 in the main text) is made up of two elements. These elements are scaled by two coefficients, $\gamma_t$ and $\lambda_t$. Here, we provide an explicit formula for these coefficients. We found that our proposed scaling works better than common procedures used in the literature. For consistency with baseline methods, we also used our proposed scaling approach for DPS and TDS (which corresponds to setting $s = 0$), since these methods apply a similar update rule. We tried to use it for other baselines but it did not work well for them.

Let $\mathbf{g}_t^1 := \nabla_{\mathbf{z}_t}||\mathbf{y}_0 - \mathcal{A}(\mathcal{D}(\bar{\mathbf{z}}_0(\mathbf{z}_t)))||_2^2$, and $\mathbf{g}_t^2 := \nabla_{\boldsymbol{\mu}_\theta(\mathbf{z}_{t+1}, t+1)}||\mathbf{y}_t - \mathcal{A}(\mathcal{D}(\boldsymbol{\mu}_\theta(\mathbf{z}_{t+1}, t+1)))||_2^2$. We set the scaling coefficients $\gamma_t$ and $\lambda_t$ according to the following scheme: $\gamma_t = \mathbb{1}_{[t \geq s]} \cdot \kappa_1 \cdot \frac{1}{max(||\mathbf{g}_t^1||_2^2, 1)} + \mathbb{1}_{[t < s]} \cdot \kappa_2 \cdot (1 - \rho) \cdot \frac{1}{max(||\mathbf{g}_t^1||_2^2, 1)}$, and $\lambda_t = \mathbb{1}_{[t < s]} \cdot \kappa_2 \cdot \rho \cdot \frac{1}{max(||\mathbf{g}_t^2||_2^2, 1)}$, where $s, \rho, \kappa_1, \kappa_2$ are all hyper-parameters.

# E. Computational Cost

We add here a comparison between the compared methods in average run-time (seconds) and memory (GB) over 10 trials for sampling a single image on ImageNet box inpainting task. For TDS and PFLD we use 5 particles. For LD-SMC, we inspect several variants that differ in the number of particles and Gibbs iterations. The results are shown in Table 4. From the table, the run-time is roughly linear in the number of particles and Gibbs iterations. Yet, importantly, it can be controlled by the practitioner to trade off performance (which can be good with one Gibbs iteration and one particle) and computational demand. In addition, we note that our code is not properly optimized and improvements can be made to it.

Table 4: Average run time and memory demand on ImageNet box inpainting task.

| Method | Run Time (Sec.) | Memory (GB) |
|---|---|---|
| LDPS | 105.5 | 8.123 |
| LTDS | 418.5 | 19.86 |
| ReSample | 333.4 | 5.769 |
| PSLD | 144.5 | 9.473 |
| PFLD | 469.8 | 26.51 |
| LatentDAPS | 67.82 | 4.745 |
| LD-SMC (1 particle) | 136.3 | 9.213 |
| LD-SMC (3 particles) | 375.1 | 15.11 |
| LD-SMC (5 particles) | 537.2 | 21.16 |
| LD-SMC (10 particles) | 1013. | 35.78 |
| LD-SMC (1 particle; 2 Gibbs iterations) | 271.2 | 9.213 |
| LD-SMC (1 particle; 4 Gibbs iterations) | 541.0 | 9.213 |

# F. Further Ablation Studies

**Number of Gibbs iterations.** Recall that in the main text we used only one Gibbs step throughout. Here we examine the effect of using more Gibbs steps in Figure 6. We do so on two tasks, the first is super-resolution and the second in Gaussian deblurring, both on ImageNet. In super-resolution, we observe an improvement in the PSNR with the number of Gibbs steps while suffering from reduction in the FID. Interestingly, this experiment shows that we can obtain a comparable PSNR to ReSample while having a substantially better FID compared to it on this task. In Gaussian deblurring there is a less clear trend, nevertheless, the figure shows that the FID can be improved while maintaining roughly the same PSNR value. We speculate that the difference between these two cases stems from the different $s$ values used in these experiments. Taking $s > 0$ tends to generate smoother images that are better aligned with higher values of distortion metrics such as PSNR.

**Proposal distributions.** In Table 5 we compare different choices of proposal distributions. Specifically, we compare to (1) the naive choice of using the prior as a proposal distribution, i.e., a bootstrap filter (Gordon et al., 1993); and (2) taking the DPS step as a proposal distribution, which corresponds to setting $s = 0$. The table shows a clear advantage to the proposal distribution used in this study and presented in Section 4.2.2.

Table 5: *FFHQ*. Proposal distribution ablation. Box inpainting on 1024 test examples using $N = 5$ particles.

| | Perceptual Quality | | Distortion | | |
|---|---|---|---|---|---|
| | FID ($\downarrow$) | NIQE ($\downarrow$) | PSNR ($\uparrow$) | SSIM ($\uparrow$) | LPIPS ($\downarrow$) |
| Prior proposal | 62.35 | 7.628 | 12.60 | 0.365 | 0.671 |
| DPS proposal | 39.94 | 7.558 | **24.20** | 0.814 | 0.235 |
| LD-SMC | **33.87** | **7.066** | 24.10 | **0.821** | **0.211** |

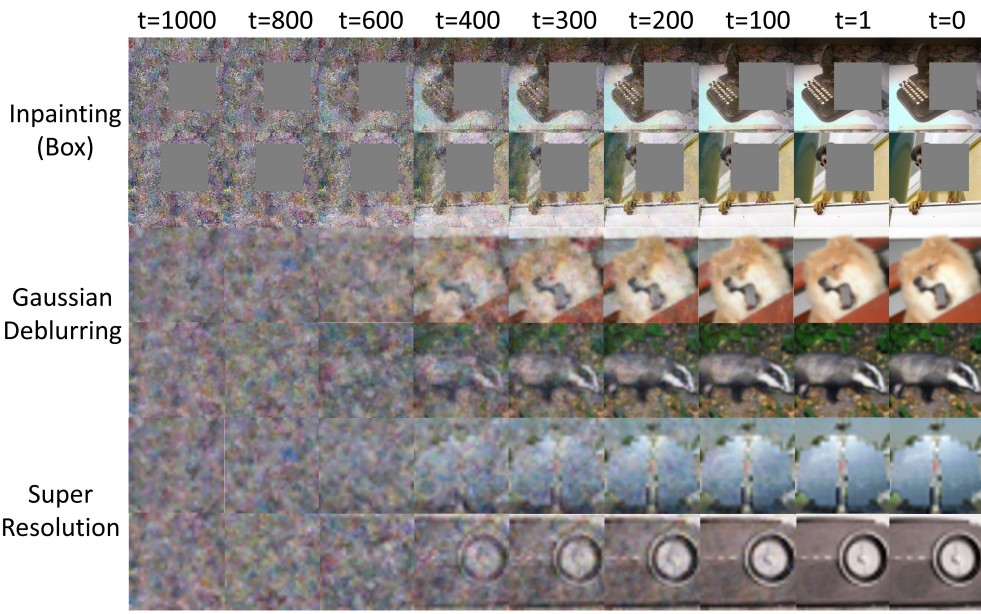

Figure 7: Evolution of $\mathbf{y}_t$ over time for different tasks according to forward process of DDIM.

## G. Forward Process

In Figure 7 we present the evolution of the auxiliary labels $\mathbf{y}_t$ over time as part of the forward process according to our proposed sampling procedure in Section 4.2, steps (a) & (b). From the figure, we observe a gradual cleaning of noise in the auxiliary labels when advancing from time $t = 1000$ to time $t = 0$.

## H. Full Results

Table 6: *ImageNet*. Box inpainting on 1024 test examples.

| | Perceptual Quality | | Distortion | | |
|---|---|---|---|---|---|
| | FID ($\downarrow$) | NIQE ($\downarrow$) | PSNR ($\uparrow$) | SSIM ($\uparrow$) | LPIPS ($\downarrow$) |
| LDPS | 65.04 | 7.935 | 19.51 | 0.665 | 0.379 |
| LTDS | 64.74 | 7.907 | 19.49 | 0.665 | 0.378 |
| ReSample | 90.32 | 8.464 | 18.16 | 0.695 | **0.318** |
| PSLD | 71.15 | 8.042 | 18.10 | 0.565 | 0.434 |
| PFLD | 72.83 | 7.933 | 18.09 | 0.564 | 0.436 |
| LatentDAPS | 98.24 | 11.36 | **19.98** | **0.704** | 0.394 |
| LD-SMC (1 particle) | 51.49 | **6.878** | 19.38 | 0.672 | 0.326 |
| LD-SMC (5 particles) | **50.67** | 6.891 | 19.42 | 0.672 | 0.325 |

Table 7: *ImageNet*. Free-form inpainting on 1024 test examples.

| | Perceptual Quality | | Distortion | | |
|---|---|---|---|---|---|
| | FID (↓) | NIQE (↓) | PSNR (↑) | SSIM (↑) | LPIPS (↓) |
| LDPS | 53.47 | 7.867 | 23.76 | 0.722 | 0.334 |
| LTDS | 52.75 | 7.884 | 23.74 | 0.721 | 0.334 |
| ReSample | 44.15 | 7.104 | 22.78 | **0.778** | **0.248** |
| PSLD | 62.38 | 8.037 | 21.65 | 0.610 | 0.411 |
| PFLD | 63.34 | 8.026 | 21.59 | 0.609 | 0.414 |
| LatentDAPS | 68.65 | 9.625 | **24.75** | 0.766 | 0.330 |
| LD-SMC (1 particle) | 38.21 | 6.969 | 23.54 | 0.727 | 0.285 |
| LD-SMC (5 particles) | **36.18** | **6.671** | 23.43 | 0.727 | 0.278 |

Table 8: *ImageNet*. Gaussian deblurring on 1024 test examples.

| | Perceptual Quality | | Distortion | | |
|---|---|---|---|---|---|
| | FID (↓) | NIQE (↓) | PSNR (↑) | SSIM (↑) | LPIPS (↓) |
| LDPS | 52.48 | 6.855 | 23.61 | 0.615 | 0.383 |
| LTDS | 50.82 | 6.695 | 23.57 | 0.614 | 0.379 |
| ReSample | **46.45** | 7.411 | 24.36 | 0.639 | **0.353** |
| PSLD | 60.68 | **6.599** | 21.94 | 0.506 | 0.417 |
| PFLD | 60.94 | 6.733 | 21.78 | 0.496 | 0.421 |
| LatentDAPS | 77.09 | 10.55 | **24.44** | **0.659** | 0.417 |
| LD-SMC (1 particle) | 52.48 | 6.855 | 23.61 | 0.615 | 0.383 |
| LD-SMC (5 particles) | 52.29 | 6.791 | 23.61 | 0.615 | 0.383 |

Table 9: *ImageNet*. Super-resolution (8×) on 1024 test examples.

| | Perceptual Quality | | Distortion | | |
|---|---|---|---|---|---|
| | FID (↓) | NIQE (↓) | PSNR (↑) | SSIM (↑) | LPIPS (↓) |
| LDPS | 61.02 | 6.514 | 21.65 | 0.523 | 0.439 |
| LTDS | 59.12 | 6.270 | 21.59 | 0.520 | 0.435 |
| ReSample | 87.65 | 8.290 | 22.05 | 0.532 | 0.491 |
| PSLD | 66.56 | 7.669 | 20.83 | 0.480 | 0.489 |
| PFLD | 64.72 | 7.685 | 20.83 | 0.479 | 0.492 |
| LatentDAPS | 104.6 | 12.68 | **22.38** | **0.566** | 0.489 |
| LD-SMC (1 particle) | 58.06 | 6.243 | 21.51 | 0.524 | 0.434 |
| LD-SMC (5 particles) | **57.89** | **6.238** | 21.50 | 0.520 | **0.433** |

Table 10: *FFHQ*. Box inpainting on 1024 test examples.

| | Perceptual Quality | | Distortion | | |
|---|---|---|---|---|---|
| | FID (↓) | NIQE (↓) | PSNR (↑) | SSIM (↑) | LPIPS (↓) |
| LDPS | 39.81 | 7.592 | 24.15 | 0.814 | 0.236 |
| LTDS | 39.57 | 7.602 | 24.24 | 0.814 | 0.236 |
| ReSample | 86.79 | 7.142 | 19.75 | 0.815 | 0.230 |
| PSLD | 39.68 | 6.544 | 22.31 | 0.774 | 0.246 |
| PFLD | 39.06 | **6.509** | 22.40 | 0.774 | 0.245 |
| LatentDAPS | 60.24 | 9.999 | **24.91** | **0.838** | 0.257 |
| LD-SMC (1 particle) | **33.37** | 7.032 | 23.98 | 0.819 | 0.212 |
| LD-SMC (5 particles) | 33.87 | 7.066 | 24.10 | 0.821 | **0.211** |

Table 11: *FFHQ*. Free-form inpainting on 1024 test examples.

| | Perceptual Quality | | Distortion | | |
|---|---|---|---|---|---|
| | FID (↓) | NIQE (↓) | PSNR (↑) | SSIM (↑) | LPIPS (↓) |
| LDPS | 40.17 | 7.609 | 27.95 | 0.858 | 0.212 |
| LTDS | 39.78 | 7.578 | 27.95 | 0.858 | 0.212 |
| ReSample | 37.01 | **6.622** | 26.31 | **0.891** | **0.151** |
| PSLD | 36.26 | 6.835 | 26.19 | 0.823 | 0.216 |
| PFLD | 36.43 | 6.817 | 26.35 | 0.825 | 0.215 |
| LatentDAPS | 54.40 | 8.766 | **29.23** | 0.883 | 0.223 |
| LD-SMC (1 particle) | 33.67 | 7.034 | 27.35 | 0.859 | 0.194 |
| LD-SMC (5 particles) | **33.60** | 7.021 | 27.35 | 0.859 | 0.194 |

Table 12: *FFHQ*. Gaussian deblurring on 1024 test examples.

| | Perceptual Quality | | Distortion | | |
|---|---|---|---|---|---|
| | FID (↓) | NIQE (↓) | PSNR (↑) | SSIM (↑) | LPIPS (↓) |
| LDPS | 29.30 | 6.538 | 28.03 | 0.775 | 0.237 |
| LTDS | 30.23 | 6.553 | 27.93 | 0.772 | 0.238 |
| ReSample | 39.80 | 7.441 | 28.45 | 0.763 | 0.275 |
| PSLD | 36.31 | 6.802 | 24.02 | 0.633 | 0.341 |
| PFLD | 37.16 | 6.751 | 23.96 | 0.628 | 0.343 |
| LatentDAPS | 54.28 | 9.496 | **29.70** | **0.831** | 0.283 |
| LD-SMC (1 particle) | **29.19** | 6.575 | 28.37 | 0.789 | **0.232** |
| LD-SMC (5 particles) | 29.47 | **6.528** | 28.34 | 0.787 | 0.233 |

Table 13: *FFHQ*. Super-resolution (8×) on 1024 test examples.

| | Perceptual Quality | | Distortion | | |
|---|---|---|---|---|---|
| | FID (↓) | NIQE (↓) | PSNR (↑) | SSIM (↑) | LPIPS (↓) |
| LDPS | **29.64** | **6.412** | 25.48 | 0.701 | 0.282 |
| LTDS | 30.45 | **6.412** | 25.38 | 0.698 | 0.284 |
| ReSample | 59.23 | 7.307 | 25.55 | 0.661 | 0.356 |
| PSLD | 40.33 | 6.803 | 23.66 | 0.615 | 0.347 |
| PFLD | 38.11 | 6.832 | 23.69 | 0.617 | 0.345 |
| LatentDAPS | 70.24 | 10.17 | **26.87** | **0.760** | 0.344 |
| LD-SMC (1 particle) | 30.02 | 6.426 | 25.52 | 0.706 | **0.277** |
| LD-SMC (5 particles) | 30.62 | 6.455 | 25.49 | 0.703 | 0.278 |

# I. Additional Image Reconstructions

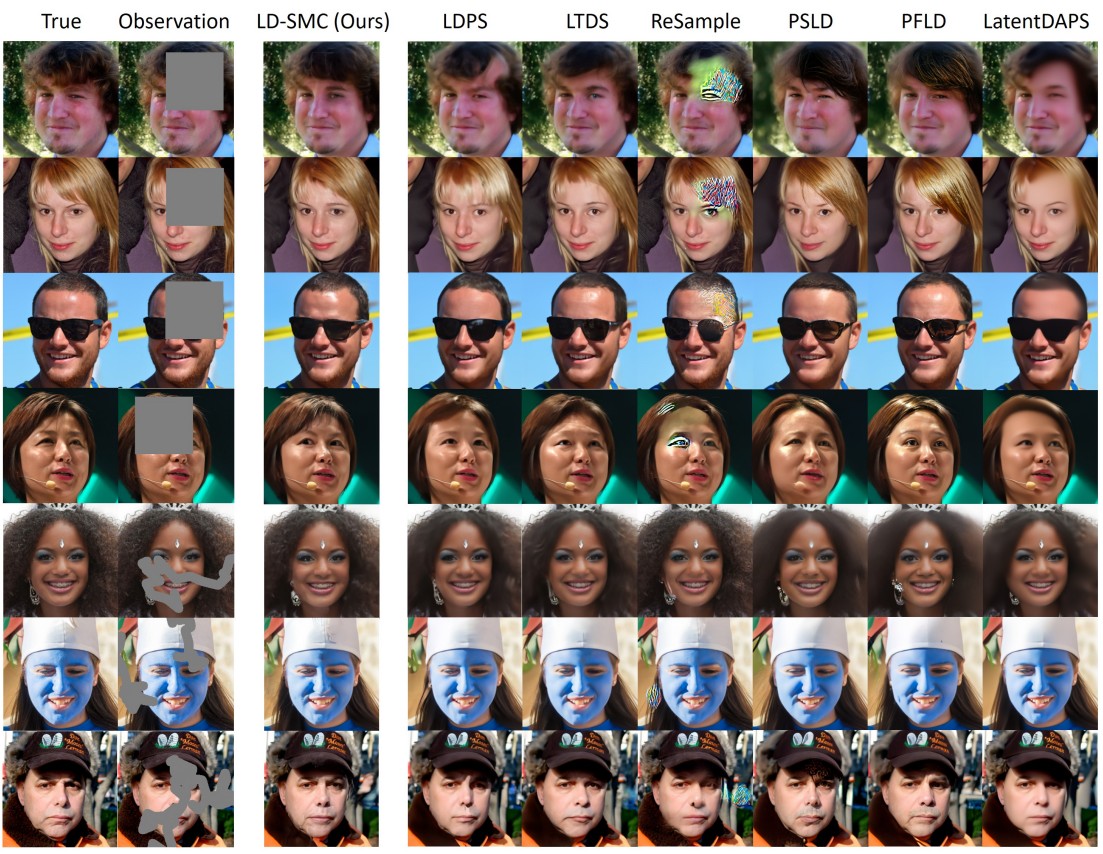

Figure 8: Comparison between LD-SMC and baseline methods on inpainting of FFHQ images.

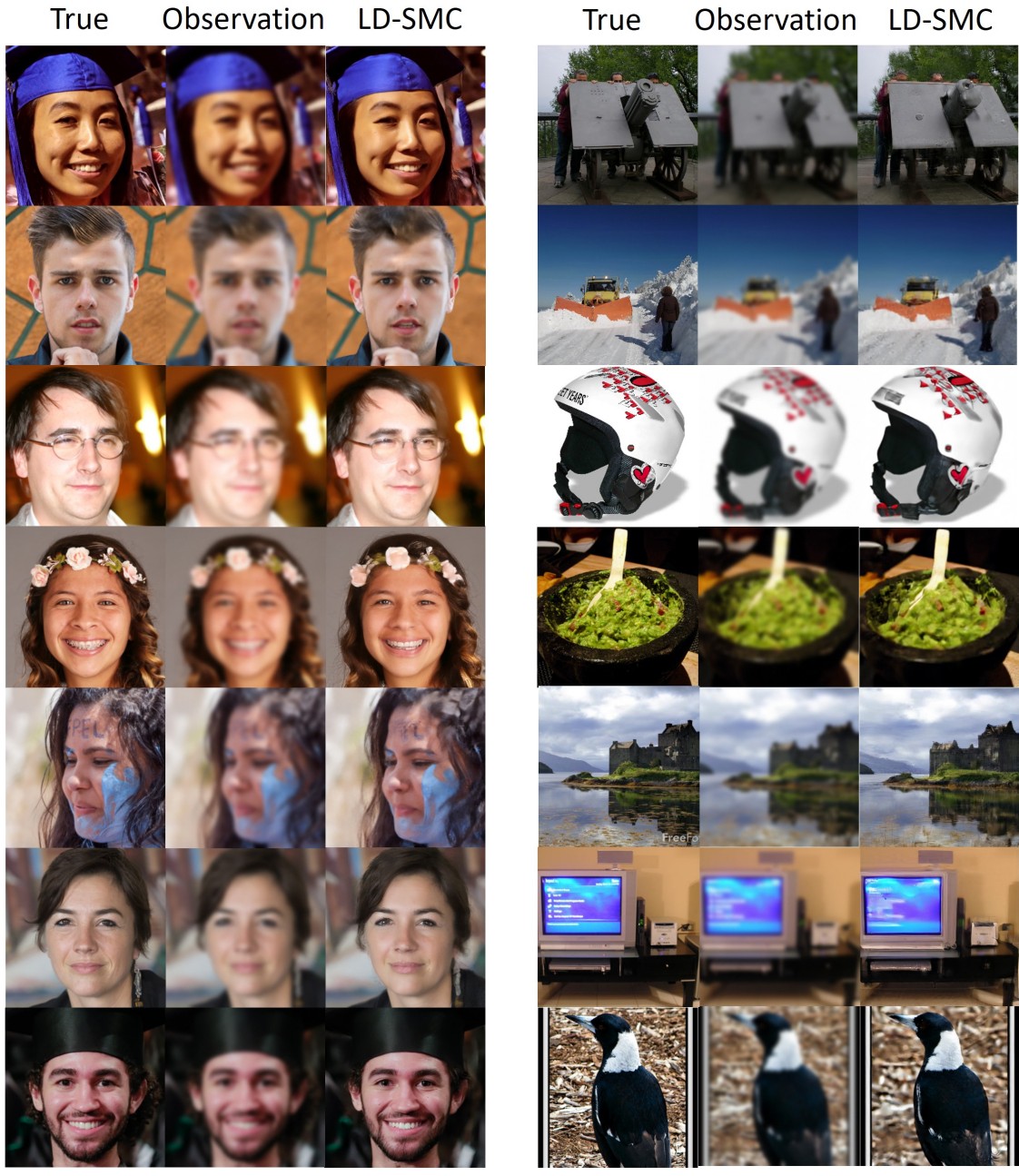

Figure 9: *Gaussian deblurring*. LD-SMC reconstruction of images from FFHQ (left) and ImageNet (right).

True    Observation    LD-SMC        True    Observation    LD-SMC

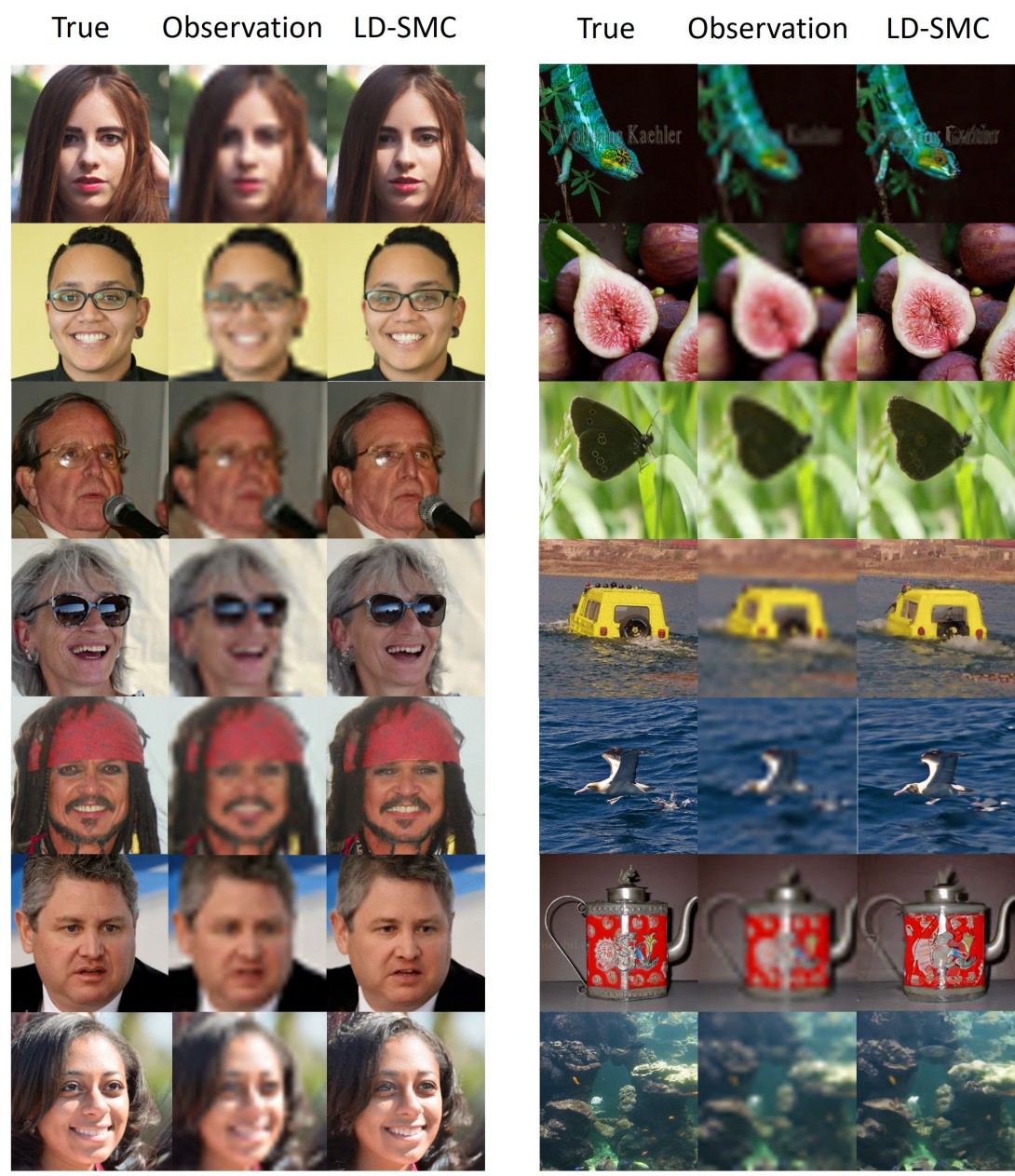

Figure 10: *Super-resolution*. LD-SMC reconstruction of images from FFHQ (left) and ImageNet (right).

