# OpenReview forum: "Inverse Problem Sampling in Latent Space Using Sequential Monte Carlo"
_ICML.cc/2025/Conference — ICML 2025 poster_

### Official Review · Reviewer_326R · 2025-03-12

**Overall Recommendation:** 4

**Summary:**

This paper is about solving inverse problems in a training-free manner using latent denoising diffusion priors and sequential Monte Carlo methods. It proposes a novel probabilistic models that enables inference of the hidden states with latent diffusion priors given the observation.

Specifically, this probabilistic models relies on combining two "orthogonal" previous ideas introduced in the literature. On one hand, a popular approximation for the likelihood of the observation, of which the gradient is required to be able to implement a diffusion posterior model given the noisy state is $p(y|x_t) \approx p(y|E(x_0|x_t))$. Another approach consists instead of sampling drawing noisy observations according to the forward process and then conditioning, using various heuristics, the reverse process on these observations.

This paper essentially combines these two approaches and blends them in a SMC scheme. This is done by defining a specific probabilistic models in which there are hidden states of which we have observations. Given a hidden state $z_t$, the observation $y_t$ is assumed to be obtained by first decoding the hidden state $z_t$, applying the forward operator to it then adding Gaussian noise with std $\tau$. The proposed algorithm is then a SMC applied to this probabilistic model.

**Claims And Evidence:**

The main claim in the paper is that adding auxiliary observations should in combination with the DPS approximation should help in capturing the "large scale semantics" as well as the "finer details". While this is demonstrated through quantitatively through superior performance with various metrics, it is not really studied qualitatively. Can the authors exhibit an example in which this can be understood precisely?

**Essential References Not Discussed:**

While the algorithms to which the method is compared to are still relevant, there are nonentheless recent works from last year that achieve much better performance on posterior sampling with latent diffusion.

[1] Zhang, B., Chu, W., Berner, J., Meng, C., Anandkumar, A. and Song, Y., 2024. Improving diffusion inverse problem solving with decoupled noise annealing.
[2] Moufad, B., Janati, Y., Bedin, L., Durmus, A., Douc, R., Moulines, E. and Olsson, J., 2024. Variational Diffusion Posterior Sampling with Midpoint Guidance.

**Experimental Designs Or Analyses:**

- The method is tested on the standard imaging benchmark that is used in most diffusion posterior sampling papers. The method is also compared to standard latent diffusion posterior sampling methods. The analysis of the result is sound.

- The images generated with the various algorithms are much blurier than one would expect. For example, posterior sampling with pixel-space diffusion yields images with much better quality. I believe that a discussion of this matter is warranted.

**Methods And Evaluation Criteria:**

Besides some issues with the methodology that can be fixed and which I discuss below, there are some design choices that are rather odd and are not explained in the paper. For example, in the probabilistic model introduced, the authors assume that the observation is obtained by applying the forward operator to the decoding of the current noisy state. Doesn't this introduce some stability issues that could yield to blurry images? The decoder has been trained on clean data only and evaluating it at noisy samples should result in odd behavior. Can the authors further discuss this? I would have assumed that a more reasonable approach would be to evaluate the decoder at the denoised hidden state. In this case one could set the variance to be $1 - \bar\alpha_t$ instead of letting it be a hyperparameter.  Similarly, the decoder is applied on highly noisy samples in the proposal transition.

Also, the Gaussian likelihood approximation of $p(y_0|x_t)$ has variance $1 - \bar\alpha_t$. This is quite odd since $y_0$ is the clean data, why would one assume such a large variance? In Wu et al. 2024 the variance is that of the p(y_0 | x_0), could you explain what you mean by "the variance term is taken to be the variance of the forward diffusion process?

**Other Comments Or Suggestions:**

see above.

**Other Strengths And Weaknesses:**

no further strengths and weaknesses.

**Questions For Authors:**

no further questions, see above.

**Relation To Broader Scientific Literature:**

The paper borrows various ideas from existing diffusion posterior sampling methods such as [1,2,3]. The methodology presented in the paper is still interesting and is not merely a simple combination of these papers.

[1] Dou, Z. and Song, Y. Diffusion posterior sampling for linear inverse problem solving: A filtering perspective.
[2] Trippe, B. L., Yim, J., Tischer, D., Broderick, T., Baker, D., Barzilay, R., and Jaakkola, T. Diffusion probabilistic modeling of protein backbones in 3d for the motif-scaffolding problem.
[3] Song, B., Kwon, S. M., Zhang, Z., Hu, X., Qu, Q., and Shen, L. Solving inverse problems with latent diffusion models via hard data consistency.

**Theoretical Claims:**

While the idea is interesting and the methodology sounds reasonable, I believe that there are several issues with it that needs to be clarified/addressed.
- First, in the probabilistic model considered (defined in 4.1) the joint distribution of the hidden states is **not** necessarily Markovian due to the fact that the authors use DDIM. Unless $\eta = 1$, the hidden process is not Markovian. Still, the authors assume that it is Markovian as evidenced the computation that starts at line 240 in the second column. Unless I am missing something, this derivation does not hold unless $\eta=1$.
- Regarding the same derivation, in SMC one only needs to propagate the particle approximation of $p(z_{t:T} | y_{t:T})$ but here the authors instead perform what is known as *online smoothing*; they derive particle approximations of the hidden states conditioned on all the remaining observations. This is arguably harder than filtering and it seems to me that this is not needed. Actually, the authors assume that
$$
p(z_{t:T} | y_{0:T-1}) \approx p(z_{t:T} | y_{t:T-1}, y_0)
$$
which significantly simplifies the computations. Still this requires computing $p(y_0|z_t)$ but this is approximated using the DPS approximation. It seems to me that smoothing is considered here because the authors want to further condition on the initial observation.  Now I would like to emphasize that the probabilistic model considered in the paper is **arbitrary**, meaning that the observation likelihood could have been different. Hence, since the authors want to condition on both $y_0$ and $y_t$ at each step of the diffusion process they could have changed their probabilistic model so that there are **two** observations $(y^1 _t, y^2 _t)$ for the same hidden state $z_t$. If their likelihood is given by
$$
p(y^1 _t, y^2 _t | z_t) = \mathcal{N}(y^1 _t | \mathcal{A}\big(\mathcal{D}(\overline{z}_0(z_t)), (1 - \overline\alpha_t) \big)  \mathcal{N}(y_t | \mathcal{A}\big(\mathcal{D}(z_t)), \tau^2 \big)
$$
A classical particle filter with this model, with observations $y^1 _t = y_0$ for all $t$, and observations $y^2 _t$ generated as in the paper would yield, I believe, the exact same algorithm if all the steps are performed conditionally on $y^1 _{0:T}$, which is not restrictive at all. The catch however is that the observation model for $y^1 _t$ is misspecified and the targeted marginal distribution is no longer the posterior distribution of interest.

- The authors mention that the proposed algorithm is a blocked Gibbs sampling procedure but I am unsure that this is the case. A blocked Gibbs sampler in this case would proceed by first sampling the hidden states given the observations then the observations given the hidden states. The proposed algorithm however has one more step where the hidden states $z_{1:T}$ are sampled conditionally on $\hat{z}_0$ only and not the observation. I do not see how this can be a Gibbs sampler. Shouldn't one draw a whole trajectory from the particle smoothing approximation then use it to draw the observations?

---

> ### Author Rebuttal · Authors · 2025-03-31
>
> We thank the reviewer for the time and effort put into the review. We are encouraged that the reviewer found our approach interesting and the analysis sound. We address the reviewer’s comments in the following. All answers and suggestions will be incorporated in the paper.
>
> **(1)**
> > “The main claim in the paper [...] is not really studied qualitatively”
>
> The main claim in this paper is that combining auxiliary variables with the DPS approximation can perform better sampling for inverse problems. This point is supported by our experiments. The “large scale” vs “fine detail” is our qualitative observation on LD-SMC performance, and not the main claim itself. An illustrative example of this can be seen at https://tinyurl.com/2u6mmj2x, specifically the generated writing in the third image.
>
> **(2)**
> > Behaviour of the decoder and stability issues
>
> Thank you for raising this great question. We agree with the reviewer that evaluating the decoder on noisy latents, especially for large $t$ values, can generate non-natural images. However, it does not imply stability issues as long as the gradients of the decoder are well-behaved, which is verified empirically. Importantly, the latents, even if noisy, carry information about the label $\mathbf{y}_0$ which guides the sampling process using the auxiliary labels closer to the desired image. Please see also comments (4) & (7) to reviewer yJi6 that touched this point.
>
> **(3)**
> > Evaluating the decoder at the denoised hidden state
>
> Thank you for this suggestion. In our early experiments, we tried using DPS, for generating the labels and initialization of LD-SMC. Since it did not improve the results we used the proposed simple alternative which is also computationally cheaper.
>
> **(4)**
> >What do you mean by "the variance term is taken to be the variance of the forward diffusion process?”
>
> The variance of the conditional distribution $p(\mathbf{z}_t|\mathbf{z}_0)$ which is $1−\bar{\alpha}_t$.
>
> **(5)**
> > Large variance of the Gaussian likelihood
>
> The variance aims to reflect the stochasticity in $\mathbf{z}_0 | \mathbf{z}_t$, it starts small and gets larger with $t$, but capped at 1. This choice is sensible although other options are also applicable. Either way, the weighting mechanism can fix for that approximation.
>
> **(6)**
> > “In the probabilistic model [...] the authors assume that it is Markovian”
>
> The forward process is indeed not Markovian, but, in the backward process, $\mathbf{z}_t$ depends only on $\mathbf{z}\_{t+1}$ as evident in Eq. 10 & 12 in the DDIM paper.
>
> **(7)**
> > Methodology - the SMC procedure and Gibbs sampling
>
> We thank the reviewer for the valuable and important comments on LD-SMC methodology. Following the reviewer’s comments we show here that the empirical distribution over samples converges to the true target of interest $p_\theta(\mathbf{z}_0 | \mathbf{y}_0)$ in the large compute limit. This addition leads to a few modifications to the algorithm presented in the paper. Importantly, since the Gibbs sampling process was applied only once, the modifications from the current presentation are minor. In addition, we stress that these modifications preserve the results witnessed in the paper (and even slightly improve them). Link to algorithm: https://tinyurl.com/2t9rpf3j. The main changes are, (1) Generate the auxiliary label $\mathbf{y}_T$, and use it in the SMC procedure for correcting the initial sampling step at time $t=T$; (2) Sample the full chain $\mathbf{z}\_{0:T}$ using SMC, for the Gibbs sampling procedure. These modifications allow us to show the following result,
>
> Theorem (informal). Let $\mathbb{P}_N(\mathbf{z}\_{0:T}) = \sum\_{i=1}^N w\^{(i)}_0 \delta\_{\mathbf{z}\_{0:T}\^{(i)}}(\mathbf{z}\_{0:T})$, be the discrete measure obtained by the function $\mathbf{SISR}$ in Algorithm 1, where $\delta$ is the Dirac measure. Under regularity conditions $\mathbb{P}_N(\mathbf{z}\_{0:T})$ converges setwise to $p\_\theta(\mathbf{z}\_{0:T} | \mathbf{y}\_{0:T})$ as $N \rightarrow \infty$. Furthermore, the stationary distribution of the Gibbs sampling process is $p\_\theta(\mathbf{z}\_{0:T}, \mathbf{y}\_{1:T} | \mathbf{y}_0)$, and $p\_\theta(\mathbf{z}_0 | \mathbf{y}_0)$ is the limiting distribution of the $\mathbf{z}_0$'s subchain.
>
> Link to the full claim and proof: https://tinyurl.com/3rdvpnad. Importantly, this result does not depend on *any* of the approximations made to derive our model.
>
> **(8)**
> > Recent works from last year
>
> Thank you for referencing us to these studies. We will discuss them in the paper. We added here a comparison to LatentDAPS (Zhang et al., 2024) under our experimental setup. Please see the results and discussion in comment (3) to reviewer yJi6.
>
> **(9)**
> >The images generated are blurry
>
> The quality of images heavily depends on the prior diffusion model. We used LDM VQ-4 which is not as strong as recently released diffusion models. This point was also shown in Table 1 (and discussed in the appendix) of Zhang et al. (2024).

---

> > ### Comment · Reviewer_326R · 2025-04-06
> >
> > I would like to thank the authors  for their rebuttal.
> >
> > If I have understood correctly your modifications, you have:
> > - changed the probabilistic model to the one I've proposed in my review and then used a particle filter (instead of approximating a particle smoother) to obtain the particle approximation
> > - removed the steps in which you sample $z\_{1:T}$ conditionally on $\hat{z}\_0$. Now you simply draw the observations conditionally on the resampled trajectory $z\_{0:T}$.
> > - These changes yield a principled SMC-within-Gibbs algorithm and thus, theoretical guarantees, as you've provided in your link.
> >
> > I welcome these changes and now think that the methodological part is in a better shape! My only minor concern now is that I think that saying $p_\theta(z_0 | y_0)$ is the limiting distribution is a bit misleading since the reader might assume that the distribution in question is $\propto p(y|z_0) p_\theta(z_0)$, which is not the case. The limiting distribution is the marginal obtained by integrating over $y_{1:T}$ and $z_{1:T}$ and this distribution is not the same as $\propto p(y|z_0) p_\theta(z_0)$.
> >
> > > "The forward process is indeed not Markovian [...]"
> >
> > I still don't agree with the argument here. My point is that in DDIM the joint distribution
> > $$
> > p\_0(z\_0) p(z\_T | z\_0) \prod_{t = 1}^{T-1} p(z_t | z_{t+1}, z_0)
> > $$
> > is not necessarily a Markov chain (unless you take for example the parameterization in eq. (16) of the DDIM paper with $\eta = 1$) and hence cannot be written as a backward Markov chain starting at $T$. Hence, the probabilistic model defined at the beginning of section 4.1 is **not** the same as the one to which you apply the particle filter (after having plugged the parametric approximation).
> >
> > This is not a major issue and can be fixed by directly defining the probabilistic model backwards, i.e. with $z_{t-1} | z_t \sim \int p(z_{t-1} | z_t, z_0) p(z_0 | z_t) dz_0$. I insist that the probabilistic model in this case is **different** from the one you define, unless $\eta = 1$ for example. And, the backward distribution I define is essentially the one approximated by DDIM.
> >
> > The additional experiments are a plus and I think they strengthen the paper. I have raised my score from 2 to 4.

---

> > > ### Author Response · Authors · 2025-04-07
> > >
> > > We are glad our additional clarifications and results successfully addressed most of your concerns. We will integrate all the changes in the paper. Thank you for your insightful and constructive feedback, and for raising the score!
> > >
> > > Please see our answers regarding your comments,
> > >
> > > > “If I have understood correctly your modifications, you have [...] ”
> > >
> > > Yes, this is correct.
> > >
> > > > “[...] The limiting distribution is the marginal obtained by integrating over $\mathbf{z}\_{0:T}$ and $\mathbf{y}\_{0:T}$ and this distribution is not the same as $\propto p(\mathbf{y}|\mathbf{z}_0)p\_θ(\mathbf{z}_0)$”
> > >
> > > We agree, we make this point precise in the full proof at the provided link. The Gibbs sampling procedure allows us to take samples from the joint distribution over $\mathbf{z}\_{0:T}, \mathbf{y}\_{1:T}\ | \mathbf{y}_0$, but ultimately we care about samples from the marginal of that distribution, namely $\mathbf{z}_0 | \mathbf{y}_0$. We will clarify this point in the main text to prevent any confusion.
> > >
> > > > “[...] the probabilistic model defined at the beginning of section 4.1 is not the same as the one to which you apply the particle filter (after having plugged the parametric approximation) [...]”
> > >
> > > Thank you, this is a great point. In DDIM the backward process is trained to mimic the non-markovian forward process. We implicitly assumed that this approximation is accurate in the generative model which we will make explicit. In addition, we will follow your suggestion and modify the generative model according to it in the paper.

---

### Official Review · Reviewer_yJi6 · 2025-03-14

**Overall Recommendation:** 2

**Summary:**

The authors propose a sampling method based on Sequential Monte Carlo (SMC) in the latent space of diffusion models. The proposed approach leverages the forward process of the diffusion model to introduce additional auxiliary variables (e.g. noisy measurements), followed by SMC sampling as part of the reverse process.

The method is evaluated on commonly used benchmarks: ImageNet and FFHQ, demonstrating improvements in Gaussian deblurring, inpainting, and super-resolution tasks. The results suggest that incorporating SMC in the latent space of diffusion models may enhance sampling efficiency and reconstruction quality at the cost of expensive gradient updates (see Eq. (4)) and hyper-parameter tuning (see Sec 4.2.3).

While the approach is promising, further clarification on the computational trade-offs and runtime analysis would strengthen the contribution. Additionally, an ablation study on the role of auxiliary variables and each gradient update from Eq. (4) in improving sample quality could provide deeper insights into the method's effectiveness.

**Claims And Evidence:**

The claims are not supported with clear evidence. Please see the weaknesses below.

**Essential References Not Discussed:**

Essential references have been discussed. However, the experimental results lack comparison with these methods.

**Experimental Designs Or Analyses:**

Yes, I have checked the soundness/validity of experimental designs or analyses presented in Section 5.

**Methods And Evaluation Criteria:**

Proposed method and evaluation criteria make sense. However, the extent of the evaluation is limited and relevant baselines in particle filtering area are missing (please see weaknesses below).

**Other Comments Or Suggestions:**

Please see the weaknesses above.

**Other Strengths And Weaknesses:**

### Strengths

The experimental results demonstrate improvements over prior works in Gaussian deblurring, super-resolution, and inpainting tasks. These findings suggest that the Sequential Monte-Carlo (SMC) approach effectively enhances posterior sample quality in diffusion models.

### Weaknesses

1. Given that the proposed method employs a multi-particle sampler, it would be valuable to compare its performance against existing particle filtering methods discussed in related works, such as PFLD and FPS. Although the compared baselines are relevant, a direct comparison with these particle-based methods could provide insights into the advantages and potential limitations of the proposed approach, particularly in terms of sampling efficiency, particle diversity, and robustness in diffusion-based inverse tasks.

2. Why is the initial guess for $\hat{z}_0$ (Sec 4.2.1) as the argmin of Eq (3) important? Can’t we just use the original image itself? Especially, because the VAE-encoder will produce a valid latent if the input is an image. The solution of the optimization problem in Eq (3) on the other hand may not produce an image.

3. The main idea of noising the measurements and performing a gradient step to minimize the measurement error in the noisy latent space has been previously explored (see for example ADIR: https://arxiv.org/pdf/2212.03221). Additionally, the multi-particle sampling approaches have also been extensively studied in prior works, as discussed in the related works of this paper (Sec 3). Given these prior works, the contribution of the proposed approach appears limited.

4. Line 243 (right col): Why is $y_t$ independent of $y_0$ in the first equality?

5. In Eq (4), the gradient wrt $\hat{z_t}$ can be decomposed by applying chain rule with variable $\mu_{\theta}(\cdot,\cdot)$. In this case, the multiplying factor $\gamma_t$ seems to be the Jacobian of $\mu$ wrt $\hat{z}_t$. So the second and third terms are roughly on the same scale, except some scaling factors and additive constants. In that case, what is the justification for using them both? Besides, why would the third term give any meaningful gradients given the fact that $D$ is not trained to produce valid images given noisy latents?


##############
Post-rebuttal
##############

The reviewer thanks the authors for responding to the raised concerns. Based on the newly added results, the reviewer is raising the score to borderline reject. Please see below for the remaining concerns.

The reviewer has the following concerns regarding the contributions of the paper post rebuttal.

The authors claim that a paper very similar to theirs, called PFLD, is a concurrent work (although it had first appeared on ArXiv in Aug 2024). The idea of noising measurements has also previously appeared in ADIR. The authors argue that this technique can not be extended to latent diffusion models due to the non-linearity of encoder-decoder. However, this is the case for most of the compared baselines including the proposed method. For examples, $y_t$ will not provide useful signal unless the decoder preserves some structure of the original image (as shown in Figure 1) through linear approximation. Therefore, the main contribution of this paper appears limited given the fact that it combines these two ideas without properly crediting these prior works.

The argument that $y_t$ is independent of $y_0$ is also not necessarily true in general. It is a mere implication of the model structure assumed by the authors. There is no guarantee that this is the true structure. The gradient in the second term of Eq. (4) can be decomposed by applying chain rule with $mu_\theta$.

Majority of the relevant baselines were omitted during the main paper submission and only appeared during the rebuttal. Given these concerns, the reviewer believes the paper needs a major revision before getting accepted.

**Questions For Authors:**

Please see the weaknesses above.

**Relation To Broader Scientific Literature:**

The main idea of noising the measurements and performing a gradient step to minimize the measurement error in the noisy latent space has been previously explored (see for example ADIR: https://arxiv.org/pdf/2212.03221). Additionally, the multi-particle sampling approaches have also been extensively studied in prior works, as discussed in the related works of this paper (Sec 3). Given these prior works, the contribution of the proposed approach appears limited.

**Theoretical Claims:**

This is an empirical paper.

---

> ### Author Rebuttal · Authors · 2025-03-31
>
> We thank the reviewer for the time and effort put into the review. We are encouraged that the reviewer found our approach specifically and SMC in general promising. We address the reviewer’s comments in the following. All answers and suggestions will be incorporated in the paper.
>
> **(1)**
> > “Further clarification on the computational trade-offs and runtime analysis”
>
> Thank you for this valuable suggestion. Please see comment (4) to Reviewer AxT3.
>
> **(2)**
> > “The role of auxiliary variables and each gradient update from Eq. (4)”
>
> Thank you for raising this point. Please see the following figures on box inpainting that analyze that. ImageNet- https://tinyurl.com/yrht7az9; FFHQ - https://tinyurl.com/2snc4vdz. Similar to Table 3 in the paper the figures show the tradeoff between FID and PSNR when varying $s$, the hyper-parameter that balances between the terms. Taking $s=0$ is equivalent to using only the first gradient term which leads to a large degradation in the FID. Conversely taking $s$ to be too large results in a steep degradation in the PSNR between $s=333$ and $s=500$. Using $s=333$ strikes a good balance between the FID and PSNR in inpainting tasks.
>
> **(3)**
> > Contribution and relation to prior works
>
> Thank you for referring us to ADIR, we will address it in the paper. Similar to FPS, ADIR noise the measurements based on the assumption of a linear corruption operator. As such, it is not readily clear how to use these methods with latent diffusion models due to the decoder's non-linearity. A main contribution of our paper is the ability to use auxiliary labels in latent diffusion models. The auxiliary labels are then used as an integral part of our SMC procedure.
>
> Regarding particle-based sampling approaches. As far as we know there are only five relevant studies, MCGDiff (Cardoso et al., 2023), FPS (Dou & Song, 2024), TDS (Wu et al., 2024),  SMCDiff (Tripper et al., 2023), and PFLD  (Nazemi et al., 2024). Specifically, MCGDiff and FPS both rely on the assumption of a linear corruption operator and cannot be used with latent diffusion models due to the non-linearity of the encoder-decoder. TDS is being extensively compared against. As for SMCDiff, it was designed mainly for motif-scaffolding, and not general inverse problems. Nonetheless, the proposal used in that study, namely the prior diffusion model, can be used in our case as well. PFLD is a concurrent study that uses PSLD update as a proposal distribution, which we did compare to in the paper.
>
> Following the suggestions of this reviewer and reviewer 326R, we add here a comparison to PFLD, LD-SMC with the prior as a proposal distribution, and LatentDAPS (Zhang et al., 2024). The comparison is presented in Tables 1 & 2 at https://tinyurl.com/p6hh6bjy. Per comment (7) to Reviewer 326R, please note that there are small changes from the results reported in the paper for LD-SMC. From the tables, LD-SMC significantly outperforms PFLD and LD-SMC with the prior proposal in all metrics. In comparison to LatentDAPS, LD-SMC has a clear advantage in FID, NIQE, and LPIPS, while LatentDAPS is better in PSNR and SSIM.
>
> **(4)**
> > Initial guess for $\mathbf{\hat{z}}_0$
>
> Thank you for raising this interesting question. Indeed there are multiple valid ways to initialize $\mathbf{\hat{z}}_0$. We did not use the original image since in some tasks (such as super-resolution) it cannot be applied due to a mismatch in the image dimensions. The important part of the initialization is that it will carry information about the measurement $\mathbf{y}_0$ which can then be used to guide the sampling process using the auxiliary labels. Fig. 5 in the paper shows that $\mathbf{y}\_{1:T}$ are sensible. Tables 9 & 10 at https://tinyurl.com/p6hh6bjy show an advantage for LD-SMC initialization compared to reviewer’s proposal, perhaps because it  doesn’t take into account the corruption operator.
>
> **(5)**
> >”Why is $\mathbf{y}_t$ independent of $\mathbf{y}_0$?”
>
> Due to the model structure when conditioning on $\mathbf{z}_t$ these variables become conditionally independent.
>
>
> **(6)**
> > ”In Eq. (4) [...] what is the justification for using them both?”
>
> We would appreciate further clarification in case we misunderstood the question. The terms use different labels and we do not see how these gradients or their scale are the same. Moreover, $\gamma_t$ is a free scalar parameter and it doesn’t have to be connected to the Jacobean.
>
> **(7)**
> >”Why would the third term give any meaningful gradients [...]?”
>
> Thank you for raising this question. As the decoder wasn’t trained on noisy images, we do not expect its reconstructions to be natural images, especially for large $t$ values. However, as a fixed function, its gradients are meaningful even on out-of-distribution data as they have information on how this function reacts to small changes. As such, the gradients help LD-SMC to move the latent encoding to one whose reconstruction is closer to the desired image as verified empirically.

---

### Official Review · Reviewer_AxT3 · 2025-03-14

**Overall Recommendation:** 4

**Summary:**

The work proposed a Sequential Monte Carlo based sampling algorithm for solving imaging inverse problems with latent diffusion model. The writing is clear and easy to follow.

**Claims And Evidence:**

The claim is solid. Despite achieving the highest perceptual quality, this work suffers from a significant loss in distortion quality. This critical trade-off needs to be explicitly addressed in the main paper (moving distortion quality into the tables in the main paper). Failure to do so would be misleading.

**Essential References Not Discussed:**

It has discussed most of the important works.

**Experimental Designs Or Analyses:**

Yes, the comparison is fair and well-designed.

**Methods And Evaluation Criteria:**

Yes, the evaluation is fair and well-designed. To help better understanding the computational cost, please provide a computational cost comparison with baselines methods including memory and inference time per image.

**Other Comments Or Suggestions:**

No more.

**Other Strengths And Weaknesses:**

Weakness:
(1) This work presents a method that builds upon existing techniques, notably TDS, by extending the sampling process to the latent space, with a method introduced in Resample baseline. While this extension successfully introduces Sequential Monte Carlo to the latent space and demonstrates performance improvements, it comes at a significant computational cost. The reliance on multiple particle sampling and decoder operations during guidance presents a practical challenge.

**Questions For Authors:**

(1) The number of particles used in the proposed method is 5, is there any reason for choose this number?  An ablation study and a corresponding discussion will be helpful to inform if scaling up the particles can further improve the performance or not.
(2) By using the weighting and resampling method as introduced in this paper, I think there will be a improvement in the worst-case performance, which may be the main reason for the improvement in the table. I am wondering if authors can add a experiments or discussion to cover this part.

**Relation To Broader Scientific Literature:**

No.

**Theoretical Claims:**

There is no theoretical claims.

---

> ### Author Rebuttal · Authors · 2025-03-31
>
> We thank the reviewer for the time and effort put into the review. We are encouraged that the reviewer appreciated LD-SMC perceptual quality and the evaluation part. We address the reviewer’s comments in the following. All answers and suggestions will be incorporated in the paper.
>
> **(1)**
> > “This work suffers from a significant loss in distortion quality.”
>
> Across 24 comparisons of distortion metrics (including free-form inpainting in comment (2) to Reviewer sS3Y), LD-SMC is first 3 times, second 10 times, third 10 times, and fourth one time. Hence, in our view, LD-SMC is comparable to baseline methods in distortion metrics.
> In general, there can be a trade-off between metrics that represent perceptual quality and those that represent distortion (Blau & Michaeli, 2018). As mentioned in the paper, we gave a stronger emphasis to the perceptual metrics, since the goal, as we see it, is to obtain high-quality images. Conversely, we could have put more emphasis on distortion metrics, for example when performing a hyper-parameter search, but it would have come at the expense of the perceptual quality.
>
> **(2)**
> > Moving distortion metrics to the main text
>
> Thank you for the suggestion. Due to lack of space, we report in the main text the FID and NIQE which are considered perceptual metrics, and LPIPS, a distortion metric. Note that it is common in the literature to report only the FID and LPIPS in the main tables, (e.g., Chung et al., 2023b; Dou & Song 2024). Nevertheless, we will include all the metrics in the next version of the paper.
>
> **(3)**
> > “The number of particles used in the proposed method is 5, is there any reason for choosing this number?”
>
> Thank you for this suggestion. We chose 5 particles because of run-time and memory considerations. In SMC one can expect an improvement in the performance when increasing the number of particles, yet as the reviewer mentioned, in the latent space it can be costly due to encoder-decoder operations. Following the reviewer's suggestion, Table 6 in https://tinyurl.com/p6hh6bjy shows an ablation for the number of particles. The table shows a general trend of improvements in perceptual and distortion metrics when increasing the number of particles. Importantly, even taking only one particle results in favorable performance for LD-SMC compared to baseline methods at a similar computational cost. Per comment (7) to Reviewer 326R, please note that there are small changes from the results reported in the paper for LD-SMC.
>
> In addition, we examine the performance of LD-SMC using one particle and multiple Gibbs iterations in Tables 7 & 8 at https://tinyurl.com/p6hh6bjy. From the tables, results can be improved even when using one particle (which reduces the computational demand of the method) by using multiple Gibbs iterations.
>
>
> **(4)**
> > “Please provide a computational cost comparison with baselines methods including memory and inference time per image.”
>
> Thank you for this suggestion.
> We add here a comparison between the methods in average run-time (seconds) and memory (GB) over 10 trials for sampling a single image. For LD-SMC we inspect several variants, including using only 1 particle and multiple Gibbs iterations. As can be seen in comment (3), using 1 particle allows to almost match the results of LD-SMC with 5 particles in inpainting tasks and when allowing to take multiple Gibbs iterations the performance can be further enhanced in Gaussian debluring and super-resolution. From the table, the run-time is roughly linear in the number of particles and Gibbs iterations. Yet, importantly, it can be controlled by the practitioner to trade off performance (which can be good with one Gibbs iteration and one particle) and computational demand. In addition, please note that our code is not properly optimized and we believe that large improvements can be made in the run time of LD-SMC.
>
> | | Run time (sec.) | Memory (GB)
> |----|---|---|
> | Latent DPS | 105.5 | 8.123
> | Latent TDS | 418.5 | 19.86
> | ReSample | 333.4 | 5.769
> | PSLD | 129.8 | 9.590
> | LD-SMC (1 particle) | 136.3 | 9.213
> | LD-SMC (3 particles) | 375.1 | 15.11
> | LD-SMC (5 particles) | 537.2 | 21.16
> | LD-SMC (10 particles) | 1013  | 35.78
> | LD-SMC (1 particle; 2 Gibbs iterations) | 271.2 | 9.213
> | LD-SMC (1 particle; 4 Gibbs iterations) | 541.0 | 9.213
>
> **(5)**
> > “By using the weighting and resampling [...] an improvement in the worst-case performance, [...] I am wondering if authors can add an experiment or discussion to cover this part.”
>
> Thank you for this valuable feedback. Indeed the weighting and resampling correct for the approximations in the proposal distribution and posterior (section 4.2.2). We formalize this claim in comment (7) to Reviewer 326R.

---

> > ### Comment · Reviewer_AxT3 · 2025-04-01
> >
> > Dear authors,
> >
> > I appreciate your response. You've effectively resolved all my questions. I will raise my score to accept (4).

---

> > > ### Author Response · Authors · 2025-04-03
> > >
> > > We are glad that our additional clarifications and results successfully addressed your concerns. We will integrate all the changes in the paper. Thank you for raising the score and for your valuable and supportive feedback!

---

### Official Review · Reviewer_sS3Y · 2025-03-14

**Overall Recommendation:** 2

**Summary:**

This paper studies inverse problems using sequential Monte Carlo sampling in a latent space. Generative models are great priors for the inverse problems. Although diffusion models achieve great performance, due to their computationally expensive reverse process and sequential nature, leveraging diffusion models in inverse problems raises technical challenges. To address the technical difficulty in sampling from the exact posterior distribution, the authors proposed sequential Monte Carlo-based sampling method. The authors proposed a new posterior approximation and proposal distribution. The proposed method achieves competitive performance.

**Claims And Evidence:**

Claim 1. The authors claim that the proposed method is effective in capturing both large-scale patterns and local fine-grained details.

Incorporating diffusion models into Inverse problems is technically challenging especially for the joint posterior of original image x at all time points, namely, p(x_0, … x_T | y). In the literature, recent works have attempted to address this challenge in various ways. One line of approaches based on P(Y|E[x_0|x_t]) captures large scale patterns and the other line of approaches with auxiliary y_{1:T} captures fine-grained local details. The authors claimed that the proposed method achieves the best of both. Figure 2 demonstrates the traits of two different approaches and the proposed method outperforms both lines of methods in the in-painting task. Figure 4 shows another results in deblurring. But the quantitatively is not analyzed.

**Essential References Not Discussed:**

More baselines: RED-diff, DDRM, DDNM^+ \PI GDM and so on. To make the comparison more comprehensive, the authors may want to include other diffusion model-based inverse problem methods even though they are not performed in the latent space.

**Experimental Designs Or Analyses:**

The experimental setting is pretty standard. No discussion is needed. However, more baselines can be included.

**Methods And Evaluation Criteria:**

Method. The proposed method is reasonable and is a variant of recent approaches. The sequential Monte Carlo has been proved effective in sampling with diffusion models. The approximation in Section 4.2.2. is introduced but no analysis of this approximation is provided.

Evaluation. The evaluation was performed similarly to previous works. 256 x 256 ImageNet and FFHQ 256 x 256 with 1024 samples were used. The authors have reported the performance in various evaluation metrics: FID, NIQE, LIPIPS, SSIM, PSNR. The authors mainly presented  Perceptual Quality metrics and distortion metrics (PSNR, SSIM) are presented in the appendix.

Overall, the proposed method and evaluation are reasonable.

**Other Comments Or Suggestions:**

In the main paper, if the performance in all metrics are provided in the main paper, it would be better for readers to understand the strengths and weaknesses of the proposed method.

The difference between the proposed method and existing SMC methods needs to be highlighted.

**Other Strengths And Weaknesses:**

Strengths

1. Interesting exploration. This paper studies a new sampling scheme, inspired by the prior work on sequential Monte Carlo sampling, for inverse problems with a new proposal distribution and its update.

Weaknesses

1. Weak experimental results. Although the proposed method achieves competitive performance in inpainting, its efficacy in other inverse problems is not fully proven. The method exhibit poor performance in deblurring and the proposed method did not outperform baselines, especially in distortion metrics. Also, experiments in other inverse problems are missing such as super-resolution, motion deblurring, colorization and so on.
2. No analysis on the main contributions: approximation and proposal distribution. The accuracy of approximation and efficiency of the proposal distribution (computational cost, distance to posterior distribution or gap induced by the proposal distribution, the number of rejections if rejection sampling schemes are utilized.) are not discussed.

**Questions For Authors:**

Q. Any quantitative results to separately evaluate the quality of restored images in large patterns and local details. Regarding distortion metric, PSNR, SSIM, LPIPS in Table 4 and other Tables in the supplement, Resample is overall comparable or often outperforms LD-SMC. Please provide more explanation.

Q. Analysis of proposal distribution? If a proposal distribution is closer to the posterior distribution and easy to draw sample, then the proposal distribution is ideal. The effect of proposal distribution update is reported in Table 3. But this kind of thorough analysis is not presented. Also, compared to other simple proposal distributions, the performance gain is not fully analyzed.

Q. Similar to other methods, computational costs (the number of parameters, latency) are not compared. Does the proposed method induce additional computational overhead?

Q. Why does the proposed method exhibit weak performance in Gaussian Deblurring?

**Relation To Broader Scientific Literature:**

No relation to any findings in scientific literature is explicitly discussed by authors and I did not observed either.

**Theoretical Claims:**

No theoretical results are presented.

---

> ### Author Rebuttal · Authors · 2025-03-31
>
> We thank the reviewer for the time and effort put into the review. We are encouraged that the reviewer regarded our approach as *Interesting exploration* and overall appreciated it. We address the reviewer’s comments in the following. All answers and suggestions will be incorporated in the paper.
>
> **(1)**
> > ”More baselines: RED-diff, DDRM, DDNM^+ \PI GDM [...]”
>
> Thank you for pointing us to these methods. We will make sure to reference them in the paper. These baselines were not included since the values reported in them are based on a different diffusion model and hence are not comparable. As such, due to computational limitations, we picked the set of recent works that were closest to our approach and could be applied with latent diffusion, and we ran all of the baselines ourselves under the exact same experimental setup. Please see comment (3) to reviewer yJi6 for additional baselines added here in the rebuttal.
>
>
> **(2)**
> >”Weak experimental results [...] poor performance in deblurring [...] especially in distortion metrics. Also, experiments in other inverse problems are missing [...]”
>
> Thank you for the comment, but we believe otherwise regarding the results of LD-SMC. First, we would like to stress that there can be a trade-off between metrics that represent perceptual quality and those that represent distortion (Blau & Michaeli, 2018). We gave a stronger emphasis to the perceptual metrics, since the goal, as we see it, is to obtain high-quality images. Second, in image generation, the metrics can be biased, and visual inspection should be taken into account as well.
>
> While we did not achieve SoTA results on Gaussian deblurring, visually though most methods yield good and comparable reconstructions. This makes the distinction between methods on Gaussian deblurring very nuanced, and it is not obvious whether the metrics used are sufficient to capture the differences between all methods. We, therefore, focused our efforts on the more challenging inpainting task where LD-SMC significantly outperforms baseline methods, both visually and in terms of perceptual metrics.
>
> Regarding additional tasks, we present results for super-resolution in the paper. In addition, we add here results for free-form inpainting on ImageNet and FFHQ following the protocol suggested by Saharia et al., (2022). For numerical results please see Tables 3 & 4 in https://tinyurl.com/p6hh6bjy. Qualitative examples can be seen at https://tinyurl.com/2u6mmj2x
> From the tables, LD-SMC outperforms baseline methods in terms of perceptual metrics and is comparable to baseline methods in terms of distortion metrics. Visually, as in box inpainting tasks, LD-SMC reconstructions better preserve fine details compared to baseline methods. In addition, despite having good values in several metrics, ReSample presents artifacts that make the images look non-natural.
>
>
> **(3)**
> > “No analysis on the main contributions: approximation and proposal distribution.”
>
> Thank you for this valuable feedback. Please see comment (7) to reviewer 326R where we torch upon this point. Regarding the proposal, the theory is very general and allows a large leeway to pick an appropriate distribution. Ideally, we would want a proposal that generates samples that are in agreement with $\mathbf{y}_0$ and have a high likelihood. We experimented with numerous formulations and we picked the one that worked best.
>
> **(4)**
> > Difference from existing SMC methods
>
> Thank you for the suggestion. Please see comments (3) to Reviewer yJi6 in which we discuss other SMC methods.
>
>
> **(5)**
> > ”Any quantitative results to separately evaluate the quality of restored images in large patterns and local details.”
>
> We believe the common metrics, such as FID and NIQE reflect that. It can be witnessed visually as well.
>
> **(6)**
> > “Resample is overall comparable or often outperforms LD-SMC”
>
> In terms of perceptual metrics, LD-SMC outperforms ReSample (13/16 comparisons in favor of LD-SMC). In terms of distortion metrics, indeed ReSample has an advantage (15/24 comparisons in favor of ReSample). Visually ReSample presents significant artifacts in all tasks and especially in in-painting. An additional visual comparison on ImageNet Gaussian deblurring is here: https://tinyurl.com/54pec8kn. Zooming in on ReSample images, noticeable artifacts can be seen in the generated images.
>
> **(7)**
> > “Compared to other simple proposal distributions, the performance gain is not fully analyzed”
>
> Thank you for this suggestion. Table 5 in https://tinyurl.com/p6hh6bjy shows a comparison of LD-SMC with the proposed proposal distribution and two alternatives, DPS as a proposal, namely taking the hyper-parameter $s=0$, and the prior as a proposal distribution.
> From the table, LD-SMC proposal outperforms both alternatives in most metrics.
>
> **(8)**
> >Computational cost of LD-SMC
>
> Thank you for this suggestion, please see comment (4) to Reviewer AxT3 who also raised this point.

---

> > ### Comment · Reviewer_sS3Y · 2025-04-08
> >
> > I appreciate the authors' detailed response. The rebuttal addressed some of the concerns raised in my initial review. In particular, the authors revisited their proposal distribution and compared the Prior proposal across Tables 1 to 5 in the 'Additional Results' section on the anonymous GitHub repository. However, its empirical benefit is not fully demonstrated. Furthermore, the proposed method was not compared against alternative methods and failed to achieve competitive performance. In addition, the main contribution of this work is the extension of SMC to the latent space. Overall, the technical contributions remain somewhat limited. Therefore, I will stick to my original rating.

---

> > > ### Author Response · Authors · 2025-04-08
> > >
> > > Thank you for the additional feedback. Please see our comments below.
> > >
> > > **1. Empirical benefit of the proposal:** First please note that in Table 5 we compare our proposal distribution to the prior proposal distribution **and** the DPS as a proposal. Indeed other proposal distributions can be constructed, but this one yielded perceptually good images, especially in inpainting tasks. In addition, per comment 2 to reviewer yJi6, in figures https://tinyurl.com/yrht7az9 (ImageNet) and https://tinyurl.com/2snc4vdz (FFHQ) we analyze the effect of varying the hyper-parameter $s$. Taking $s=0$ leads to a large degradation in the FID. Conversely taking $s$ to be too large results in a steep degradation in the PSNR between $s=333$ and $s=500$. Using $s=333$ strikes a good balance between the FID and PSNR in inpainting tasks. We will be happy to perform additional comparisons to other proposal distributions according to the reviewers’ suggestions.
> > >
> > > **2. The proposed method was not compared against alternative methods:** Here in the rebuttal we added two recent baselines, PFLD and LatentDAPS. Both are suited for diffusion models in the latent space and we were able to run both under exactly the same experimental setup as LD-SMC. The comparison is shown in Tables 1 and 2 here https://tinyurl.com/p6hh6bjy. LD-SMC significantly outperforms both in FID, NIQE, and LPIPS. LatentDAPS has an advantage in PSNR and SSIM, but it still does not disqualify the merits of LD-SMC.
> > >
> > > Regarding the initial proposal of the reviewer “RED-diff, DDRM, DDNM^+ \PI GDM [...]”. Although the numbers in these papers are not comparable to ours due to mismatch in the setup (e.g., different diffusion model), we will add them and other common methods to the paper for completeness. Specifically, DDRM is designed for linear inverse problems and hence is not applicable to our use case. The other baselines can be applied with non-linear operators, but to the best of our knowledge, were not tested using latent diffusion with a decoder network, which is highly non-linear and imposes a significant computational burden. The different setup, including the decoder involvement, can impose non-trivial changes and extensive hyper-parameter search to these baselines; nevertheless, we will examine how to implement them in our case as well. Due to lack of time we cannot do it by the end of the rebuttal.
> > >
> > > **3. LD-SMC performance:** In comparison to the methods presented in the paper, in terms of the perceptual metrics, LD-SMC is first 8 times, second 5 times, and third 3 times. In distortion metrics, LD-SMC is first 4 times, second 13 times, and third 7 times. In terms of perceptual metrics, LD-SMC has a clear advantage and in terms of distortion metrics it is competitive. However, as we stated, we chose to put more emphasis on perceptual quality, which is reflected in perceptual metrics and visual inspection. Conversely, we could have performed other design choices (e.g. in the proposal distribution) or hyper-parameter tuning to favor distortion metrics, but we believe the former is more important.
> > >
> > > **4. Technical contribution:** There are several novel contributions to this work. First, we show how to combine auxiliary variables with latent space diffusion models. Second, we construct a generative model and perform inference using Gibbs sampling, of which the SMC is only part. Third, per comment 7 to reviewer 326R we theoretically show that LD-SMC is asymptotically accurate, namely that it can sample from $p\_\theta(\mathbf{z}_0 | \mathbf{y}_0)$ despite all the approximations made (link to proof: https://tinyurl.com/3rdvpnad). Lastly, we show significant improvements in perceptual quality in inpainting tasks, arguably one of the most challenging inverse problems tasks. Overall, we believe LD-SMC lays a good foundation for SMC methods for inverse problems in the latent space of diffusion models.
> > >
> > > We will be happy to provide further clarifications to the reviewer and we kindly ask from the reviewer to reevaluate our paper and the score based on our comments and the merits of our method.

---

### Decision · Program_Chairs · 2025-05-01

**Decision:**

Accept (poster)

**Comment:**

The authors introduced a new sequential Monte Carlo-based  algorithm to sample from the latentspace of diffusion models.    The main contribution is a new procedure to solve inverse problems with latent diffusion models and the use of SMC methods to solve posterior sampling problems. They initially validated their approach using two datasets (ImageNet and FFHQ). Some reviewers were concerned by the fact that the ideas of the paper were already explored in other contributions and that the contribution is incremental while the experiments do not match state-of-the-art results in some settings.

Although the paper shares some similarities with  diffusion posterior sampling methods, the methodology is still new and after the discussion period the authors provided an updated version of the algorithm which improved the empirical results. The authors also provided extensive numerical simulations, adding computation costs and ablation studies (on the number of particles for instance) during the rebuttal period. They also clarified the contributions with respect to previous works combining SMC algorithms and diffusion models.
The discussion should lead to a deep revision of the paper but I believe this would provide an interesting contribution.

For all these reasons, I lean towards accepting the paper.

Best regards.